Article 

# Characterization of paramagnetic states in an organometallic nickel hydrogen evolution electrocatalyst

Sagnik Chakrabarti[1,2], Soumalya Sinha[1,2], Giang N. Tran[1,2], Hanah Na ®[1] & Liviu M. Mirica ®[1] ✉

Significant progress has been made in the bioinorganic modeling of the paramagnetic states believed to be involved in the hydrogen redox chemistry catalyzed by [NiFe] hydrogenase. However, the characterization and isolation of intermediates involved in mononuclear Ni electrocatalysts which are reported to operate through a $Ni^{I/III}$ cycle have largely remained elusive. Herein, we report a $Ni^{II}$ complex (NCHS2)Ni(OTf)2, where NCHS2 is 3,7-dithia-1(2,6)-pyridina-5(1,3)-benzenacyclooctaphane, that is an efficient electrocatalyst for the hydrogen evolution reaction (HER) with turnover frequencies of ~3,000 $s^{-1}$ and a overpotential of 670 mV in the presence of trifluoroacetic acid. This electrocatalyst follows a hitherto unobserved HER mechanism involving C-H activation, which manifests as an inverse kinetic isotope effect for the overall hydrogen evolution reaction, and $Ni^I/Ni^{III}$ intermediates, which have been characterized by EPR spectroscopy. We further validate the possibility of the involvement of $Ni^{III}$ intermediates by the independent synthesis and characterization of organometallic $Ni^{III}$ complexes.

The production of $H_2$ with high turnover frequencies (TOFs) using earth-abundant molecular catalysts is an area of active research for replacing platinum as the pre-eminent catalyst in hydrogen evolution[1,2]. The primary inspiration for this comes from nature, where hydrogenases perform hydrogen redox reversibly with high turnovers and at low overpotentials[3]. For one class of these enzymes – the [NiFe] hydrogenases, the proposed mechanism involves reactive intermediates, termed the Ni-SI$_a$, Ni-R, Ni-L[4] and Ni-C[5] states, which control the proton and electron transfer steps (Fig. 1a). It has been proved that the Ni center is the redox-active site, switching between paramagnetic and diamagnetic states, while the Fe center remains divalent throughout catalysis.

Several structural and functional models[6,7] of the active site of [NiFe] hydrogenases have been reported. Pioneering work from Rauchfuss[8,9], Tatsumi[10]. Darensbourg[11], Ogo[12], Duboc[13] and others[14,15] have successfully captured the structural and spectroscopic signatures of the various intermediates (Fig. 1b)[16]. For mononuclear Ni complexes

(Fig. 1c), $Ni^I$ and $Ni^{III}$–H intermediates have been implied by various groups for stoichiometric and catalytic HER. However, evidence supporting both these oxidation states within the same system is lacking. In this regard, Ni complexes with porphyrins[17,18], organometallic pincer complexes[19,20], mixed thioether-phosphines[21] and azamacrocyles[22] have been reported in the literature. Recently, an elegant class of bioinspired Ni-thiolate complexes from Liaw[23] and Peters[24] were shown to form terminal $Ni^{III}$–H species, yet they were either catalytically incompetent[23] or weakly active[24]. Additionally, Shafaat et al. reported paramagnetic states in HER catalyzed by a Ni-substituted rubredoxin[25].

There is an inherent challenge that lies in designing ligands for synthetic systems which can stabilize both $Ni^I$ and $Ni^{III}$ as they have very different coordination preferences. $Ni^I$ is a soft Lewis acid and isolated complexes are usually low-coordinate and supported by π-acidic ligands[26], whereas $Ni^{III}$ has a proclivity for harder, anionic donors and exhibits higher coordination numbers[27,28]. Indeed, the possibility of accessing $Ni^I$ and $Ni^{III}$ with rigid square planar porphyrins or pincer

[1]Department of Chemistry, University of Illinois at Urbana-Champaign, 600 S. Mathews Avenue, Urbana, IL 61801, USA. [2]These authors contributed equally: Sagnik Chakrabarti, Soumalya Sinha, Giang N. Tran. ✉e-mail: mirica@illinois.edu

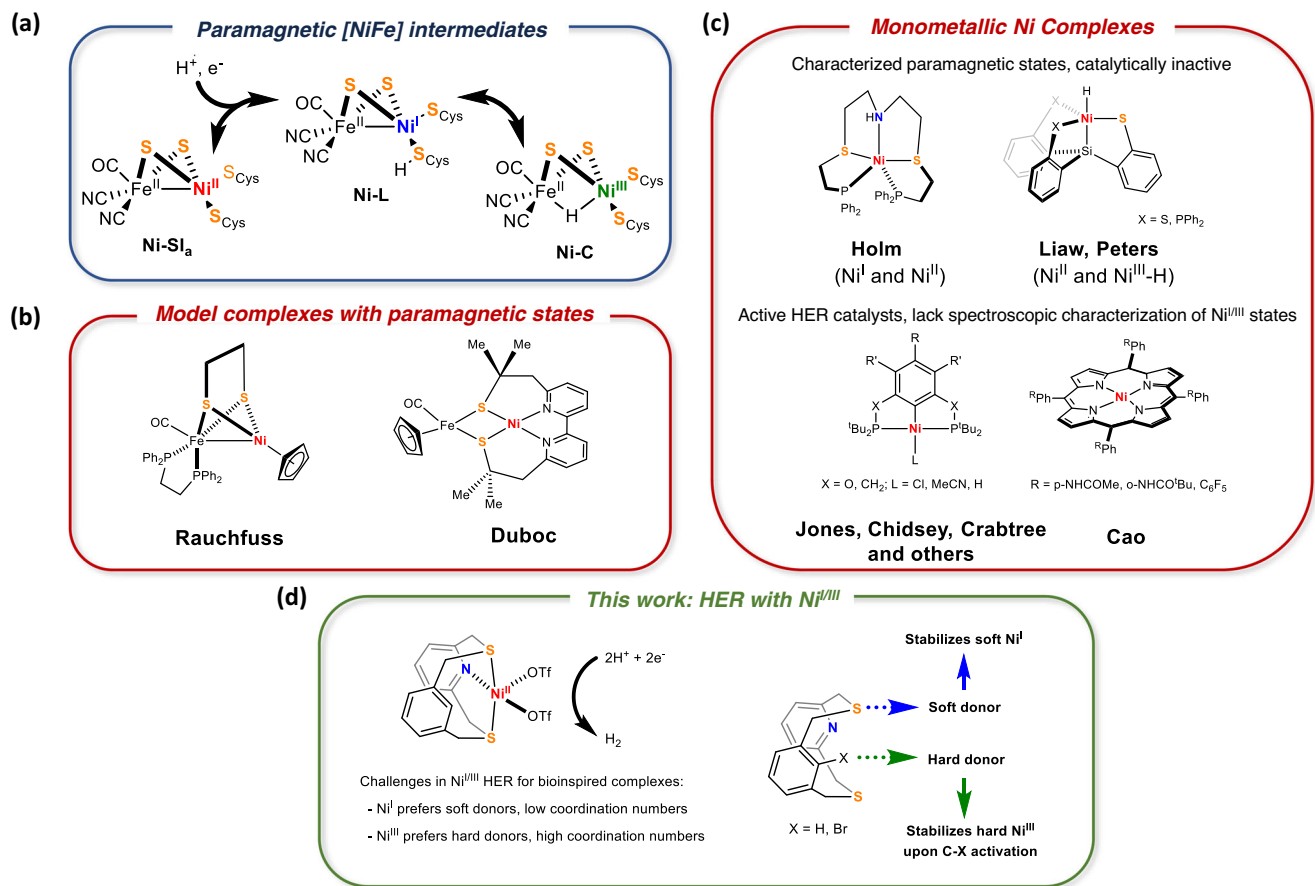

**Fig. 1 | Hydrogen evolution reaction at paramagnetic Ni. a** Paramagnetic intermediates involved in hydrogen redox catalyzed by [NiFe] hydrogenase. **b** Structural mimics of [NiFe] hydrogenase with characterized paramagnetic states and reported HER activity with strong acids. **c** Mononuclear Ni complexes with paramagnetic intermediates proposed to be involved in HER. **d** Development of a mixed hard-soft ligand enables definitive characterization of Ni-centered paramagnetic intermediates in HER.

ligands is difficult, as is evidenced by the lack of spectroscopic evidence of metal-centered paramagnetic states in the aforementioned catalysts.

Herein, we report bioinspired Ni^II complexes supported by mixed hard-soft NCHS2 and NCBrS2 ligands, which can stabilize Ni^I with soft thioether donors and allow for the isolation of well-defined Ni^III complexes when the ligands become anionic. We propose that the pre-positioning of a proximal C–H bond can be an effective strategy to generate metal hydrides that can participate in hydrogen evolution. Importantly, these Ni^II complexes were found to catalyze HER with TOFs of ~3000 s^{-1} at an overpotential of 670 mV in the presence of trifluoroacetic acid (TFA). Using a combination of synthetic, spectroscopic, electrochemical, and computational studies, we show that this new class of Ni complexes are able to access the paramagnetic Ni^I and Ni^III states during electrocatalytic hydrogen evolution. A key C–H activation step is proposed to generate a putative Ni^III–H species from a Ni^I intermediate, and we propose that it mirrors the role of the cysteine ligands in [NiFe] hydrogenases.

## Results and discussion
### Ligand Design
Recently, our group has shown that the N2S2 ligand, the dithioether analog of the N4 pyridinophane, stabilizes Pd^I complexes[29] and catalyzes HER via a Ni^{0/II} cycle[30]. We also reported that ^RN3C-type ligands allowed the stabilization of high-valent Ni centers[31–34]. In our pursuit of developing biomimetic ligands, we surmised that combining the above features may prime it for stabilizing both high-valent and low-valent oxidation states, which are invoked in the mechanism of several Ni

enzymes. This led us to synthesize the ligands NCXS2 (X = H, Br), in which the C-X bond can be activated to generate organometallic complexes. These ligand have two features which make them relevant for synthetic bioorganometallic chemistry – (a) the soft thioether donors mimic the bridging thiolates in [NiFe] hydrogenases, which acquire thioether-like properties due the strongly π-accepting CO bound to the dicationic Fe[35,36]; and (b) a carbanion donor that can stabilize high-valent metal centers.

### Synthesis and characterization of Ni^II complexes
The synthesis of **1** can be accomplished by two routes: (a) reacting (DME)NiBr2 with NCHS2 in MeCN (bond lengths and angles of NCHS2 are included in Supplementary Data 1), followed by bromide abstraction with two equivalents of AgOTf (85% yield), or (b) heating NCHS2 and Ni(OTf)2 in a MeCN/DCM mixture (65% yield). The X-ray crystallographic analysis of **1** reveals a distorted square pyramidal geometry at the Ni center (τ5 = 0.21), with one triflate ligand in the axial position, while the other triflate ligand, the N atom, and the two S atoms of NCHS2 occupy the equatorial positions (Fig. 2c, bond lengths and angles are included in Supplementary Data 2). A Ni···H–C_{aryl} interaction was observed for **1**, with Ni1–C1 and Ni1–H1 distances, 2.478 Å and 2.299 Å, respectively, and a Ni1–H1–C1 angle of 89.61°. This is similar to what was observed previously for related Ni-pyridinophane complexes, in which C_{ipso}–H bond activation was observed upon oxidation[31].

The solution magnetic moment of **1** was found to be 1.3 $\mu_B$ in MeCN, which suggests an intermediate spin state. Upon dissolving **1** in acetonitrile, a well-defined NMR spectrum was observed (Fig. 1f),

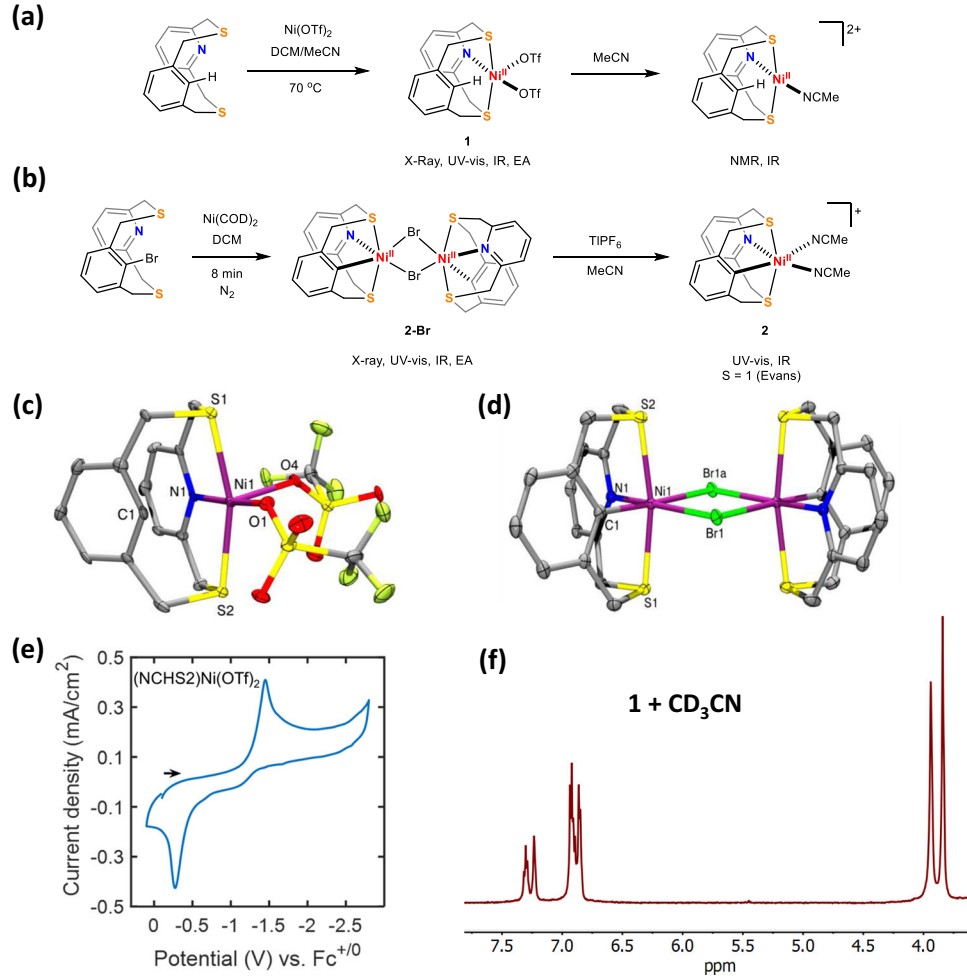

**Fig. 2 | Synthesis and characterization of Ni$^{II}$ complexes. a** Synthesis of non C–H activated complex **1** and its proposed structure on dissolution in acetonitrile. **b** Synthesis of C–H activated complexes **2-Br** and **2**. ORTEP representations (50% probability ellipsoids) for **1** (**c**) and **2-Br** (**d**). Selected bond distances for **1**: Ni1–N1 2.001(6), Ni1–O4 2.039(5), Ni1–O1 2.043(4), Ni1–S1 2.391(2), Ni1–S2 2.407(2), Ni1–C1 2.478(8), N1–H1 2.299, N1–H1–C1 89.61, and **2-Br**: Ni–N1 2.042(1), Ni–C1 1.980(1), Ni–S1 2.3759(5), Ni–S2 2.3899(5), Ni–Br1 2.5487(4), Ni–Br1a 2.720(4). **e** Cyclic voltammogram of 1 mM of **1** in 0.1 M TBAPF$_6$/MeCN in a N$_2$-saturated solution (scan rate = 0.1 V/s). **f** $^1$H NMR spectrum of **1** in CD$_3$CN showing the peaks corresponding to the aromatic and methylene protons.

and variable temperature NMR experiments show an inequivalence of the methylene protons, suggestive of a SNS planar coordination mode with an overhanging phenyl group[37]. We propose that **1** exists as a mixture of a square planar (S = 0) and square pyramidal (S = 1) species in acetonitrile, with one and two acetonitrile molecules bound respectively (see S9-S11 for a discussion on the spin state of **1**).

We also independently synthesized the organometallic complex [(NCS2)Ni(MeCN)$_2$](OTf), **2(OTf)**, via a two-step procedure and starting with the NCBrS2 ligand and Ni(COD)$_2$ (Fig. 2b). Single crystal X-ray crystallographic analysis reveals a dinuclear complex [(NCS2)Ni(μ-Br)]$_2$, which we propose yields **2** upon halide abstraction using TlPF$_6$ (Fig. 2d). The distance between the two Ni centers in [(NCS2)Ni(μ-Br)]$_2$ was found to be 3.869 Å, suggesting no significant bonding interaction between the two Ni atoms (Fig. 2d). The geometry of both Ni centers is distorted octahedral, with two two S atoms in the axial positions and the NCS2 ligand binding in a κ$^4$ conformation. The Ni–N1 and average Ni–S bond distances are 2.042 Å and 2.383 Å, respectively, while the Ni–C1 distance is significantly shorter than that in **1** (1.980 Å vs. 2.478 Å), indicating a bonding interaction.

### Electrochemical properties of Ni$^{II}$ complexes
The cyclic voltammogram (CV) of **1** under nitrogen shows a single cathodic wave at around −1.5 V vs Fc$^{+/0}$ (Fig. 1e), assigned to the Ni$^{II/I}$

reduction, and a corresponding oxidation wave at ~−0.26 V ($i_c$/$i_a$ = 1.19). No additional reduction event suggestive of a Ni$^{I/0}$ process or ligand-centered reduction was observed down to −2.75 V, implying that the Ni$^0$ is not easily accessible. We hypothesize that this large separation is indicative of an irreversible chemical process that occurs upon reducing the Ni$^{II}$ by one electron and leads to the formation of a new species that gets oxidized at a more anodic potential. As a comparison, for the Ni$^{II}$ complex with the dipyridine analog of this ligand – 3,7-dithia-1,5(2,6)-dipyridinacyclooctaphane (N2S2), two Faradaic processes corresponding to Ni$^{II/I}$ and Ni$^{I/0}$ were observed under N$_2$ (Fig. S37)[30], while there is no oxidative return wave at a more anodic potential. This implies that the substitution of pyridine with a phenyl group has a significant effect on the reductive electrochemistry of the Ni complex.

### Electrocatalytic hydrogen evolution reaction (HER) with 1
Complex **1** performed electrocatalytic HER using trifluoroacetic acid (Fig. 3a, pK$_a$ in MeCN = 12.7). The CVs collected for **1** using TFA showed catalytic peak currents with an onset potential more positive than the $E_{Ni^{II/I}}$ value and the catalytic peak potentials were at least ~500 mV more positive than that of the HER promoted by bare GC electrode in the absence of a catalyst (Fig. S23). Both complexes **1** and **2** exhibit similar CVs under similar acid concentrations (Table S1), suggesting a common pathway for HER (see pages S17–S44).

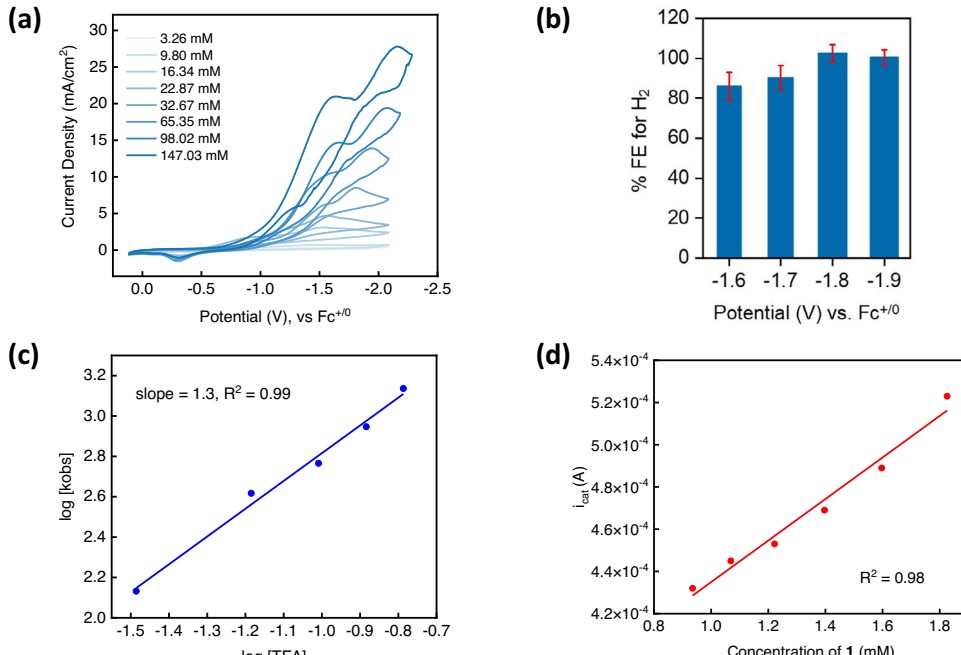

**Fig. 3 | Electrochemical hydrogen evolution with 1. a** Titrations of increasing amounts of trifluoroacetic acid in a 1 mM $N_2$-saturated MeCN solution of **1. b** Plot of variation of Faradaic Efficiencies for the production of $H_2$ on performing controlled potential electrolyses at various potentials. The error bars represent standard error and have been obtained from an average of three trials of bulk electrolysis. **c** Double-log plot of $k_{obs}$ versus TFA concentration, with a slope of 1.3, indicating a first-order reaction in acid. **d** Plot of $i_{cat}$ vs concentration of **1** obtained at 32.67 mM of TFA in a $N_2$-saturated MeCN solution.

In order to benchmark our catalyst, we measured an overpotential of 0.67 V at $E_{cat/2}$ (Fig. S49) for **1** in a 1:1 $CF_3COOH$:$CF_3COONa$ buffer, using the Appel and Helm method[38]. To calculate the $TOF_{max}$ from CVs, acid concentrations were increased for the TFA titrations. At -0.16 M TFA, scan rate independent CVs were obtained[39,40], which gives a $TOF_{max}$ of 3,016 $s^{-1}$ (Fig. S50). We also conducted electrochemical HER with added water, which showed a moderate enhancement in current densities and with a similar overpotential (Figs. S24-S26). Controlled potential electrolyses were performed at various potentials between −1.6 V to −1.9 V and high Faradaic Efficiencies (80% to 100%) were obtained for $H_2$ evolution (Fig. 3b). While Ni complexes with planar, π-conjugated ligands have a strong possibility of surface adsorption due to the better interfacing with the glassy carbon surface (as reported by Nocera et al. for Ni porphyrins)[41], we expect that the molecular geometry of **1** will disfavor strong adsorption to the electrode surface. Indeed, the homogeneous nature of the catalyst was confirmed by a rinse test (Fig. S38-S40), UV–vis spectroscopy (Fig. S45), the SEM-EDX surface analysis of the glassy carbon after the rinse test (Figs. S41–S44), and the linearity of $i_c$ and $i_p$ with $\sqrt{\nu}$ (Fig. S20)[42,43].

**Spectroscopic characterization of Ni$^I$ complexes**
The growth in catalytic current anodic of the $E_{Ni^{II/I}}$ potential suggests the involvement of Ni$^I$ in the hydrogen evolution reaction. To probe the process chemically, CoCp$^*_2$ was chosen as a reductant for an EPR-scale reduction of **1** at −35 °C. The X-band EPR spectrum of resulting solution at 77 K was simulated using two sets of rhombic g tensors (Fig. 4a, *Sim 1* and *Sim 2*) in a 4:1 ratio, respectively. We propose that the species corresponding to *Sim 1* is a four-coordinate square planar Ni$^I$ complex (**3**), with a $d_z^2$ ground state ($g_x$ ~ $g_y$ > $g_z$), which agrees with the DFT-predicted β-SOMO of **3** (Table S14, entry 2). The DFT-calculated spin density of **3** exhibits a large d-orbital character (Fig. 4d), which confirms that the ligand is not redox-active, and the reduction is metal-centered. The g-tensor for for *Sim 2* implies a $d_x{}^2 - {}_y{}^2$ ground state ($g_z$ > $g_x$ ~ $g_y$). This suggests the coexistence of two paramagnetic species with two different ground states on reducing **1**. Interestingly, the

g-values for *Sim 2* are in close agreement with those reported recently for Ni$^{III}$-H species by Peters et al.[24] and Liaw et al.[23]. We thus propose that the Ni$^I$ is poised to undergo oxidative addition with the *ipso* C–H bond of the pendant phenyl group and ascribe *Sim 2* to a C–H activated putative Ni$^{III}$-H species (**4**). DFT calculations predict that the oxidative addition is thermodynamically uphill by +0.8 kcal/mol, which suggests that these complexes will coexist in solution at room temperature. The g-anisotropy for both species is in good agreement with DFT-calculated g-values (Fig. 4c). Even though it has been shown that CW-EPR may not be able to distinguish between H and D coupling to a Ni center[24], the EPR spectrum of the reduction of **1** and (NCDS2) Ni(OTf)$_2$ with CoCp$_2^*$ showed subtle differences (Fig. S70 and Table S4), which points towards different reactivity of the Ni$^I$ center with the proximal C–H(D) bond.

A similar EPR spectrum, albeit with the two sets of g tensors in a 20:1 ratio, was obtained upon treating **2** with 1 equiv CoCp$^*_2$ at −78 °C (Fig. 4b). This experiment provides two important takeaways: (1) in the absence of the *ipso* C–H, the species corresponding to *Sim 2* was formed in significantly less amounts, (2) the reduction needed a much lower temperature. Combined, these results suggest that the presence of an *ipso* C–H bond is necessary for the formation of the species corresponding to *Sim 2* in appreciable amounts and Ni$^I$ formation is more challenging with a C–H activated Ni$^{II}$ (in accordance with HSAB theory). Addition of 1 equivalent TFA to the in situ generated Ni$^I$ complex led to the formation of a significantly simpler axial signal (Figs. S66 and S67), suggesting a complex with similar ground state as *Sim 1* and that *Sim 2* corresponds to a species which is quenched by excess protons. Indeed, electrochemical peak shift analysis also points towards an EC type mechanism operating on reducing **1** (Fig. S21). In summary, we propose that **3** and **4** exist in equilibrium in solution, mirroring the tautomeric relationship between Ni–L and Ni–C for [NiFe] hydrogenases[44].

The g-anisotropy for the proposed Ni$^I$ species corresponding to *Sim 1* is 0.24, which is remarkably close to the large g-anisotropy observed for the Ni–L state of [NiFe] hydrogenase ($g$ = [2.296, 2.118,

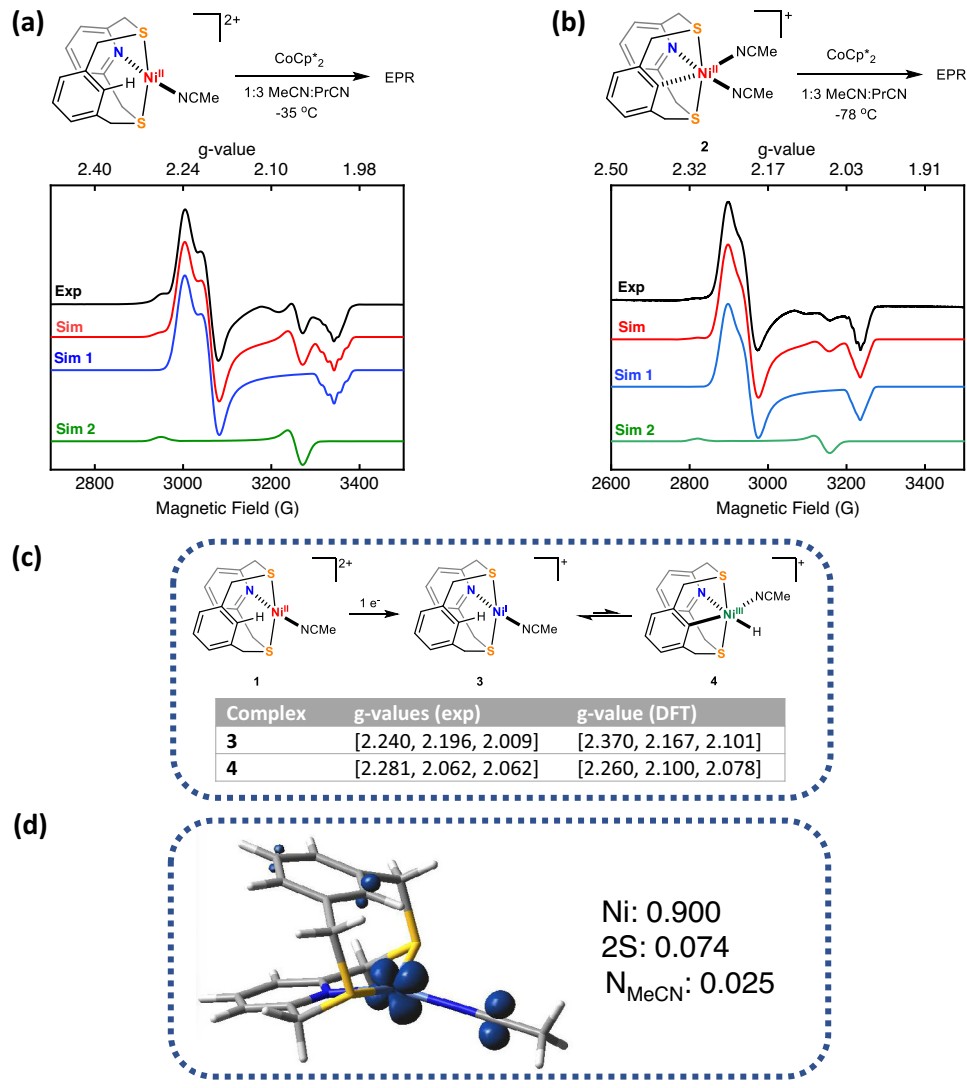

**Fig. 4 | Chemical reduction of Ni$^{II}$ complexes. a, b** Experimental (black) and simulated (red) EPR spectra for **1** and **2** after treating with 1 equiv. of Co$^{II}$Cp*$_2$ in MeCN/PrCN (1:3) at 77 K. For both cases, the simulated spectra (red) employed two sets of parameters, Sim 1 (blue) and Sim 2 (green). The following parameters were used for the simulations: **a** Sim 1: $g_1$ = 2.240 $g_2$ = 2.196, $g_3$ = 2.013 ($A_{2N}$ = 15 G), Sim 2: $g_1$ = 2.281, $g_2$ = $g_3$ = 2.062 **b** Sim 1: $g_1$ = 2.245 $g_2$ = 2.200, $g_3$ = 2.009 ($A_{2N}$ = 12 G), Sim 2:

$g_1$ = 2.305, $g_2$ = $g_3$ = 2.065. The ratios of Sim 1 to Sim 2 for (**a**) and (**b**) are 4:1 and 20:1, respectively. **c** Proposed equilibrium between **3** and **4** on the reduction of **1** and a table comparing the experimental and DFT-calculated *g*-values. **d** DFT-calculated spin density of **3** (isocontour value = 0.06), with the contributions from the relevant atoms.

2.046] with $\Delta g$ = 0.25[45]. As a comparison, most model systems which have been reported to date feature smaller values of $\Delta g$, ranging from 0.09–0.16[9,13,46]. We believe this to be a consequence of a four-coordinate Ni$^I$ center present in our system, which is also seen in the Ni-L state of [NiFe] hydrogenase. It is also worth pointing out the difference between the EPR spectrum upon the reduction of **1** with the one for the structurally-related hydrogen evolution catalyst [(N2S2) Ni$^{II}$(MeCN)$_2$]$^{2+}$ recently reported by our group[30]. The reduction of that compound generates a rhombic EPR spectrum with a g-tensor [2.205, 2.152, 2.012] that is markedly different from the one generated upon the reduction of **1**. This further supports the difference in reductive chemistry induced by the positioning of a proximal C−H bond close to the Ni center.

### Synthesis and characterization of (NCS2)Ni$^{III}$ complexes

Based on our hypothesis of a C−H activated Ni$^{III}$-H existing in equilibrium with the (NCHS2)Ni$^I$ complex, we sought to explore the possibility of stabilizing Ni$^{III}$ complexes with the NCS2 ligand system. The cyclic voltammogram of the **2-Br** shows low $E_{Ni^{II/III}}$ potentials of −0.3 V

vs Fc$^{+/0}$ and −0.46 V vs Fc$^{+/0}$ in DCM and a 4:1 DCM:MeCN mixture, respectively (Fig. 5a). This suggests and additional stability is conferred to a Ni$^{III}$ center in the presence of a coordinating solvent.

Excitingly, addition of 1 or 2 equiv AgOTf to **2-Br** in MeCN leads to the formation of the Ni$^{III}$ complexes [(NCS2)Ni$^{III}$(Br)(MeCN)]$^+$ (**[2-Br]$^+$**) and [(NCS2)Ni(MeCN)$_2$]$^{2+}$ (**[2]$^+$**), respectively, in quantitative yields (Fig. 5b). Single crystal X-ray analysis of **[2]$^+$** and **[2-Br]$^+$** reveal distorted octahedral Ni$^{III}$ centers, as expected for d$^7$ metal centers (Fig. 5c, d). The Ni−S$_{thioether}$ distances are in the range of 2.35 Å − a contraction of -0.03 Å from the corresponding Ni−S$_{thioether}$ bonds in **[2-Br]** but are in excellent agreement with the only previously reported, structurally characterized homoleptic Ni$^{III}$-thioether complex[47]. The Ni−N1 and Ni−C1 bonds are shortened by -0.05 Å vs. those in **2-Br**. To the best of our knowledge, these are the first examples of structurally characterized heteroleptic Ni$^{III}$ complexes with soft thioether donors.

Both these complexes have axial EPRs, which is suggestive of a $d_{x^2-y^2}$ ground state, with **[2-Br]$^+$** displaying superhyperfine coupling to the Br atom (Fig. 5e, f). This lends further support to our assignment of the species *Sim 2* as a Ni$^{III}$ species. Since **1** displays high rates for

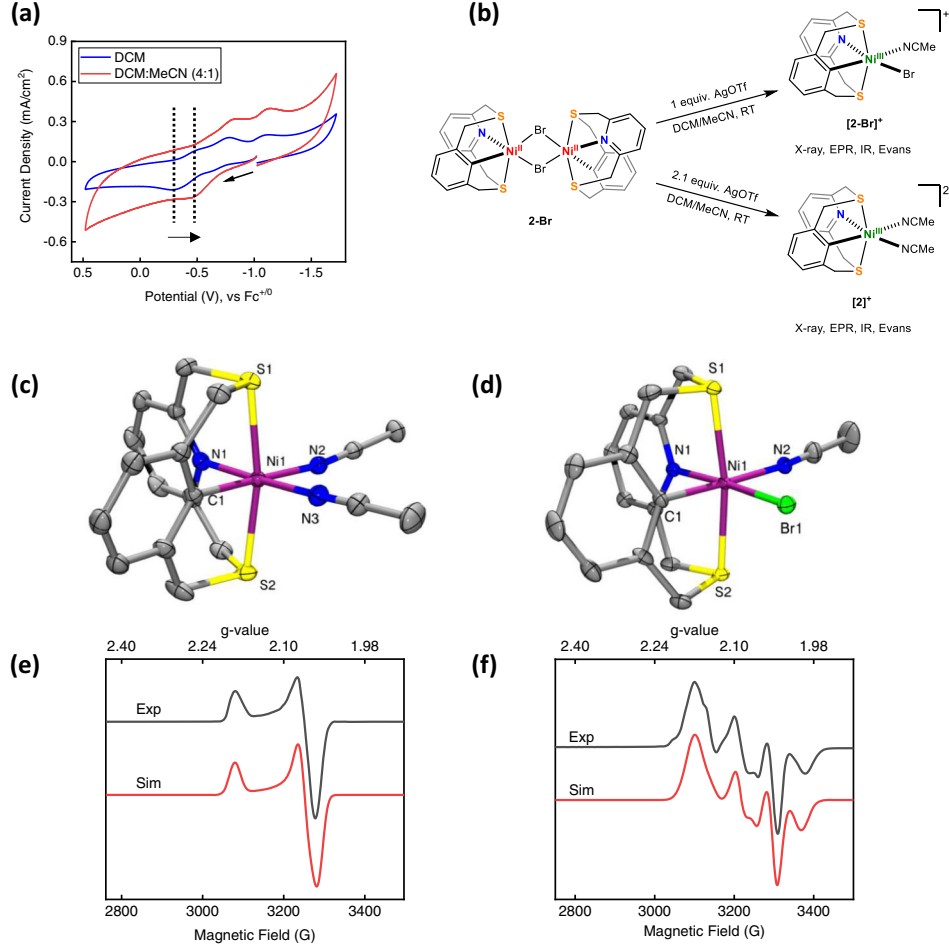

**Fig. 5 | Synthesis and characterization of C–H activated Ni^III complexes. a** Cyclic voltammogram of **[2·Br]** in dichloromethane (blue) and 1:4 acetonitrile:dichloromethane (red) showing a cathodic shift in the $E_{Ni^{II/III}}$ potential. **b** Synthesis of Ni^III complexes discussed in the text, starting from **[2·Br]**, using Ag^+ as oxidant. **c, d** ORTEP representations (50% probability ellipsoids) for **[2]^+** and **[2·Br]^+** (**c**) Selected bond distances for **[2]^+**: Ni1–S1: 2.3431(5), Ni1–S2: 2.3556(5), Ni1–N1: 1.9865(16), Ni1-N2: 1.9821(17), Ni1-N3: 1.9885(17), Ni1-C1: 1.9651(17) and (**d**) Selected

bond distances for **[2·Br]^+**: Ni1–S1: 2.3571(11), Ni1–S2: 2.3578(11), Ni1–N1: 2.024(4), Ni1-N2: 1.988(4), Ni1–C1: 1.939(4), Ni1·Br1: 2.427(5). **e, f** Experimental (black) and simulated (red) EPR spectra for **[2]^+** and **[2·Br]^+** in MeCN/PrCN (1:3) at 77 K. The simulated spectra employed the following sets of parameters, (**e**) $g_1 = 2.184$, $g_2 = 2.062$, $g_3 = 2.055$; (**f**) $g_1 = 2.164$, $g_2 = 2.064$ ($A_{Br} = 75$ G), $g_3 = 2.039$ (major species, 90%). A second paramagnetic species, possibly without bromides is simulated as a minor component (10%) with the g-tensor $g_1 = 2.192$, $g_2 = 2.064$, $g_3 = 2.094$.

hydrogen evolution, it is expected that full characterization of a reactive intermediate like Ni^III-H will be difficult. Thus, we explored other routes of generating a Ni^III–H with this ligand framework; σ-bond metathesis of B–H and Si–H bonds with Ni–Y (Y = O, F, N) species is a well-studied synthetic route to generate Ni-H complexes[48]. In fact, Liaw et al. were able to generate a transient Ni^III–H species by the metathesis of a terminal Ni^III–OPh complex with pinacolborane. Inspired by this result, we explored the possibility of generating a transient Ni^III–H species via metathesis with pinacolborane. Addition of HBPin to an in situ generated (NCS2)Ni^III phenolate species resulted in a EPR spectrum that is reminiscent of that generated upon the reduction of **1** (Fig. 4a) and suggesting the formation of the putative Ni^I–OPh and Ni^III–H species (Figs. S71-S75 and Table S5 enumerate the details of the EPR studies of the metathesis reaction).

We have thus shown that two distinct chemical processes[49], oxidative addition upon reduction to a low-valent Ni^I center and σ-bond metathesis of pinacolborane with a bona fide Ni^III-OR species, lead to a similar mix of paramagnetic species with different ground states (Table S5). Being able to access these mixtures from reduction of Ni^II and metathesis of Ni^III suggests that this system has a proclivity to undergo an oxidative addition·reductive elimination equilibrium between two different paramagnetic species. While further studies are necessary to confirm the exact chemical nature of these species, the

results presented above strongly suggest that C–H activation enables access to Ni^III species, which could include a transient Ni^III–H intermediate.

At this stage, we would like to discuss the parallels between the chemistry of **1** and [NiFe] hydrogenases. While we do not propose that the Ni compound discussed herein is an exact model of the [NiFe] hydrogenase active state, we posit a parallel between the chemistry of **1** and [NiFe] hydrogenase. In the native enzyme, the Ni^I state (Ni-L) is proposed to form a Ni^III–H species (Ni–C) by proton transfer from a protonated thiolate cysteine ligand. By comparison, in **1** this redox switch from Ni^I to Ni^III is enabled by a C–H activation step, rather than a proton transfer. Thus, we introduce a new design principle in molecular electrocatalysis, where the presence of a proximal C–H bond enables a constraint on the extent of reduction of the Ni center to Ni^I, rather than Ni^0. Such a Ni^I state is then proposed to form a Ni^III–H intermediate, which is also seen in [NiFe] hydrogenases. In contrast, the vast majority of Ni-based molecular HER catalysts involve the formation of a Ni^0 state that is required for reactivity.

### Mechanistic studies of HER

The results presented above suggest the involvement of Ni^I and Ni^III intermediates for the hydrogen evolution catalyzed by **1**. To gain

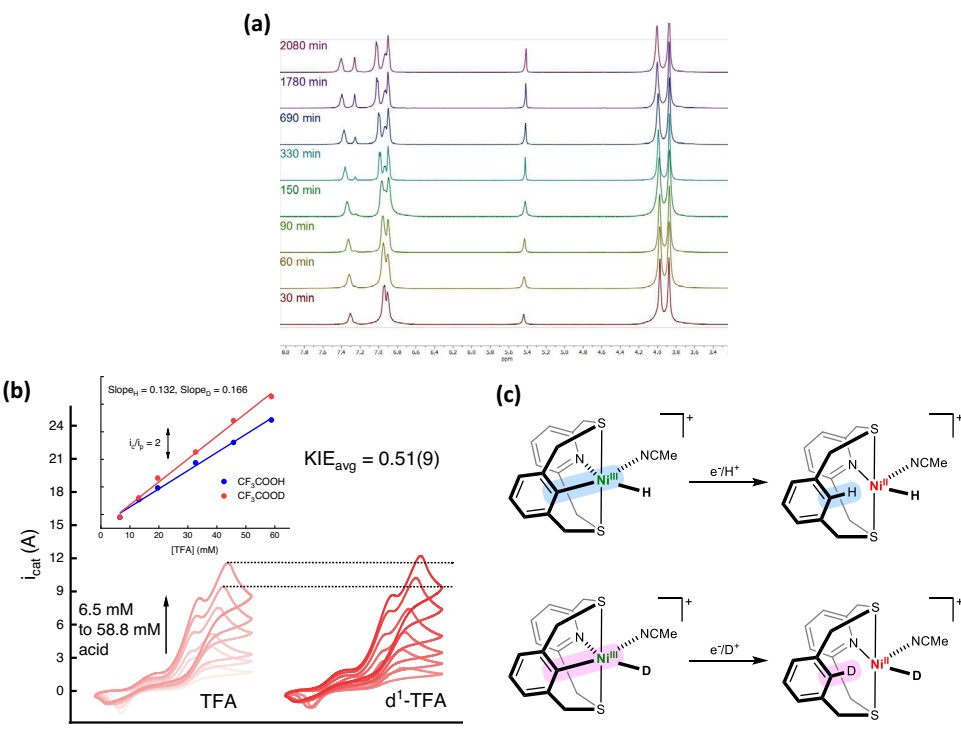

**Fig. 6 | C–H activation and electrochemical KIE measurements. a** Growth of the NMR signal corresponding to the *ipso* C–H (-7.24 ppm, singlet) bond when NCDS2 and NiOTf$_2$ were mixed together in 1:1 CD$_3$CN:CD$_2$Cl$_2$ and heated in a J-Young NMR tube and heated at 80 °C over the course of 3 days. **b** A representative example of an electrochemical kinetic isotope effect (KIE) experiment. Titrations of CF$_3$COOH or CF$_3$COOD into a 0.1 M TBAPF$_6$/MeCN produced catalytic currents. These currents were normalized by the $i_{pc}$ (Ni$^{II/I}$) under N$_2$ atmosphere at the same catalyst concentrations. The normalized currents ($i_c/i_p$) were plotted for CF$_3$COOH and CF$_3$COOD *versus* acid concentrations and the KIE was obtained as explained in the Supporting Information. **c** The likely source of inverse KIE in HER catalysis is the cleavage of the Ni–C bond to form C–H(D) bond due to a proton-coupled electron transfer step. These bonds are highlighted in blue for the Ni–H species and pink for the Ni–D species, respectively.

further insights into the mechanism, we performed several electrochemical and mechanistic studies and employed DFT calculations to support our hypotheses (*vide infra*).

### Probing the C–H activation step

A key step in the chemistry proposed above is a redox switch from Ni$^I$ to Ni$^{III}$ via a C–H activation step. In essence, we propose that C–H activation converts a weak-field ligand with soft donor atoms into a strong-field ligand with a carbanion donor, which enables the stabilization of Ni$^I$ and Ni$^{III}$ states, respectively. In this context, we have shown that the C–H activated Ni$^{III}$ compounds are directly relevant to the redox switch described above. Additional evidence for C–H activation of the NCHS2 ligand further bolsters its importance to the chemistry discussed here, and thus we have synthesized the (NCDS2)Ni$^{II}$OTf$_2$ analog to probe the C–D(H) activation at Ni$^{II}$. Excitingly, we observed that the *ipso* C–D bond of the phenyl ring exchanges with adventitious protons in a 1:1 CD$_3$CN:CD$_2$Cl$_2$ solvent mixture on heating at 80 °C over the course of 2 days (Fig. 6a). Over the course of the experiment, about 80% of the deuterium was exchanged, as tracked by using the triplet peak from the pyridine *para* C–H bond as an internal standard in the $^1$H NMR spectrum. Such a H/D exchange is only possible if there is a transient C–H activated intermediate involved. While this C–H activation is redox neutral, possibly occurring through an electrophilic mechanism at the Ni$^{II}$, this experiment proves that given a sufficient driving force the proximal C–H bond is prone to getting activated.

More direct evidence of the involvement of C–H activation during the electrochemical hydrogen evolution was also probed using ESI-MS. ESI-MS analysis of the solution from the HER electrocatalysis performed with **1** in CD$_3$CN and in presence of CF$_3$COOD/D$_2$O revealed the formation of the deuterium-labeled NCDS2 ligand (see pages S66–S69). Incorporation of deuterium in the *ipso* position of the phenyl ring of NCHS2 from the deuteroacid is only possible if a C–Ni bond is formed and cleaved by the acid during electrolysis. This further supports the activation of the C$_{ipso}$–H bond of NCHS2 during HER electrocatalysis.

### Kinetic isotope effect for HER electrocatalysis

The H/D exchange observed in the NMR experiment and deuterium incorporation in the ligand during electrolysis prompted us to investigate the electrochemical kinetic isotope effect (KIE) for **1**. An inverse KIE of 0.57 ± 0.2 and 0.51 ± 0.09 was found for E$_{cat1}$ and E$_{cat2}$ respectively from electrochemical measurements (Fig. 6b), and when using CF$_3$COOH and CF$_3$COOD as proton and deuterium sources, respectively (see pages S45–S48). If C–H activation is involved in catalysis, it is expected that there is a buildup of a species with an aryl *ipso* C–D bond when using a deuterated acid. Our calculations suggest that the conversion of **4**–**6** is the most thermodynamically uphill process and is likely involved in the rate-determining step. Such inverse KIEs have been well-known in C–H activation/formation via the intermediacy of σ-alkane complexes[50,51]. The tendency of the deuterium atom to reside in the higher frequency oscillator (C–D *versus* M–D or M–C) has been invoked to explain the inverse KIEs in those cases. While inverse KIEs have been reported for homogeneous HER systems by Gray et al.[52] and Grapperhaus et al.[53], the mechanistic implication for such an inverse effect was not offered due to the complexity of the untangling the multiple proton-electron transfer steps.

As enumerated above, the possibility of breaking a weak Ni–C bond to form a strong C–D(H) bond is the likely source of the inverse kinetic isotope effect. We propose that a proton-coupled electron

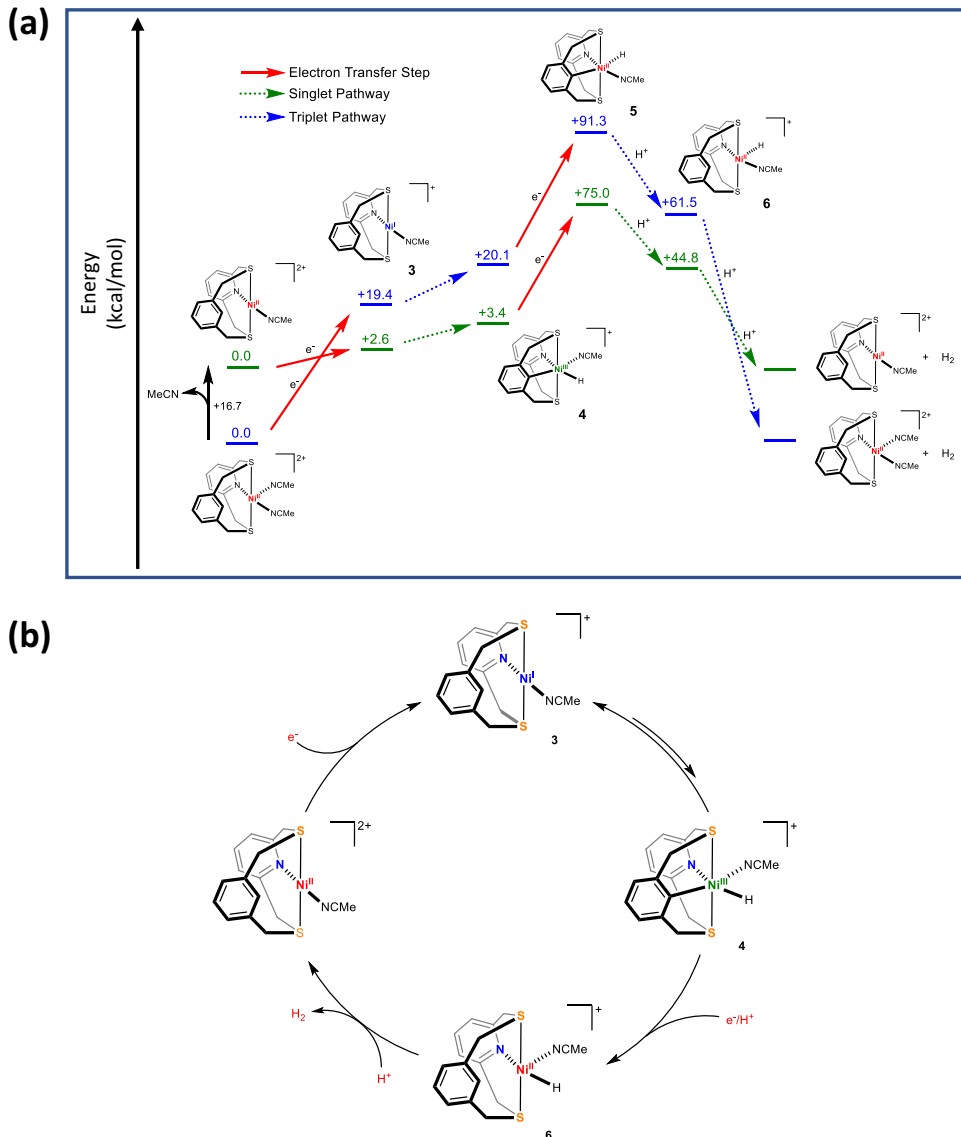

**Fig. 7 | Mechanism of hydrogen evolution reaction catalyzed by 1. a** Energy landscape of electrochemical HER catalyzed by **1**. All structures were optimized at UB3LYP/tzvp level of theory. The free energies of electron transfer reactions were calculated vs. Fc/Fc⁺ in MeCN. The free energies of proton transfer reactions were calculated by using CF₃COOH and CF₃COO⁻ in MeCN. The energy levels colored in green correspond to the S = 0 (singlet) state for Niᴵᴵ, while those colored in blue correspond to the S = 1 (triplet) state for Niᴵᴵ. The calculated $E_{cat}$ is −3.10 V for the singlet pathway and −3.08 V for the triplet pathway. The electron transfer steps are represented as red arrows. **b** Overall proposed mechanism of HER catalyzed by **1**.

transfer step is involved in converting the C–H(D) activated Niᴵᴵᴵ state to a Niᴵᴵ–H(D) where the C–H(D) bond is cleaved (Fig. 6c). This would rationalize the observation of an inverse KIE as the deuterium atom would prefer to reside in the C–D bond vs. the much weaker Ni–D bond. In order to compare the same phenomenon, we employed the related system [(N2S2)Niᴵᴵ(MeCN)₂]²⁺, where there is no possibility of C–H activation and it has been shown to operate through a Ni⁰/Niᴵᴵ·H cycle[30]. In this case, a positive KIE value of 1.17(1) was measured (Figs. S58 and S59), which further supports a different HER mechanism for **1**. It is worth noting that the electrochemical hydrogen evolution parameters observed for **1** (TOF = 3000 s⁻¹ and η = 670 mV using TFA as proton source) and [(N2S2)Niᴵᴵ(MeCN)₂]²⁺ (TOF = 1250 s⁻¹ and η = 730 mV with the same proton source) have the same order of magnitude values. Yet, the difference in the cyclic voltammograms, electrochemical kinetic isotope effect, and the EPR spectra on reducing the respective Niᴵᴵ compounds indicate that the subtle change of introducing a proximal C–H bond profoundly changes the HER mechanism at the corresponding Ni centers.

## Proposed mechanism

Since both **1** and **2** are competent electrocatalysts for the HER and they generate similar EPR spectra upon chemical reduction, we propose that they likely form the same Niᴵ intermediate **3** in solution upon electrochemical reduction. While thiolate protonation has often been invoked to facilitate the reduction of Niᴵ to Ni⁰, thioether protonation is unlikely, which combined with other electrochemical data led us to rule out the involvement of Ni⁰ in HER catalysis.

The intermediate **3** can then undergo C–H activation to form **4** (Fig. 7a). Since both structurally characterized Niᴵᴵᴵ species **[2]**⁺ and **[2-Br]**⁺ exhibit distorted octahedral geometries, we also propose an octahedral geometry for the Niᴵᴵᴵ center in **4**. We probed computationally the possibility of a concerted metalation deprotonation (CMD) at Niᴵ, followed by protonation, and the CMD step was found to be thermodynamically unfavorable by 42 kcal/mol (see page S106), and thus we propose an oxidative addition step to be the probable route to form the Niᴵᴵᴵ-H species. This would account for the large degree of irreversibility in the Niᴵᴵ/ᴵ CV feature of **1**. Since a putative Niᴵᴵᴵ-H species

is likely to be very reactive, we hypothesize that a species like [2]⁺ could form in solution by 4 reacting with adventitious protons. The independently measured CV of [2]⁺ (Fig. S32) has a quasi-reversible Ni$^{II/III}$ redox couple at −0.3 V, which is very similar to the oxidative feature seen in the CV of 1, further bolstering our hypothesis. Interestingly, [2]⁺ is also active for hydrogen evolution (Fig. S33), with CVs showing a pre-catalytic wave at −0.4 V (reduction of Ni$^{III}$ to Ni$^{II}$) and catalytic waves at potentials similar to those seen for 1.

We evaluated the possibility of heterolytic HER by protonation of 4 and it was shown to be uphill by 26 kcal/mol (see page S105 and Fig. S92) and in line with literature precedence that Ni$^{III}$−H species are usually not hydridic enough to produce H$_2$ on protonation[24]. The CVs of 1 in the presence of low concentrations of trifluoroacetic acid have two catalytic peaks (Fig. 6b), which can represent two pathways of hydrogen evolution. A possible pathway involving the heterolytic protonolysis of the Ni$^{III}$−H in 4 was considered (page S105). This step would involve the cleavage of a Ni−H(D) bond to form H$_2$ (or D$_2$) and could account for the inverse KIE, yet we consider this is less likely given the high calculated barrier for protonation of the Ni$^{III}$-H species (Fig. S92), as well as the diminished FEs observed at lower overpotentials.

The one-electron reduction of 4 to 5, a putative Ni$^{II}$-H species, has a high calculated barrier (Fig. 7a) but can be driven electrochemically. There are two possibilities for the protonation of 5 – the protonation of the Ni$^{II}$-C bond to form 6 is exergonic (Fig. 7a), whereas the protonation of the Ni$^{II}$-H bond for direct hydrogen evolution in endergonic by 22 kcal/mol (page S105). Hence, protonolysis to convert 5 to 6 is energetically favored. The overall e⁻/H⁺ step needed for the conversion of 4 to 6 can be concerted or stepwise, and we have no evidence to rule out one in favor of the other.

Finally, the protonation of 6, which is highly favorable thermodynamically based on DFT calculations, would evolve H$_2$ and regenerate 1. We propose that the mechanism shown in Fig. 7b is the dominant pathway for HER and explains the inverse KIE for E$_{cat2}$ as the conversion of 4 to 6 involves protonolysis of the C−Ni bond. Electrochemical reaction order measurements suggest a first-order reaction with respect to both the catalyst and TFA (Figs. 3c and 3d, Figs. S51–S55), and thus suggesting that homolytic HER directly from 4, 5 or 6 is not a major pathway for H$_2$ evolution.

In conclusion, we have shown that using a mixed hard-soft NCXS2 ligand (X = H, Br), it is possible to access a catalytically relevant, para-magnetic Ni$^{I}$/Ni$^{III}$ redox manifold in a Ni-based hydrogen evolution catalyst. With a combination of synthesis, EPR spectroscopy, electrochemistry, and theoretical calculations, we show for the first time that both Ni$^{I}$ and Ni$^{III}$ states are accessed during catalytic hydrogen evolution, which is directly relevant to the chemistry of [NiFe] hydrogenases. Switching between redox states by via a C−H bond activation, while commonly proposed for organometallic reactivity, is a new paradigm in small molecule electrocatalysis. Overall, we hope that this study will inspire further exploration of such mixed hard-soft donor organometallic complexes for bioinspired small molecule activation chemistry.

## Methods

### Preparation of (NCHS2)Ni(OTf)$_2$, 1

**Method 1.** To a suspension of NCHS2 (80 mg, 0.292 mmol) in MeCN was added solution of NiBr$_2$(DME) (90.30 mg, 0.292 mmol) in MeCN. Yellow solid crashed out of the solution and the reaction mixture was stirred overnight. To the suspension of NCHS2NiBr$_2$ in MeCN was added AgOTf (75.18 mg, 0.292 mmol) solution in MeCN. The color of the solution changed to light green and very light yellow solid precipitated. The solution mixture was run through a syringe filter to separate AgBr salt. A light green solid was isolated from the filtrate by adding excess amount of Et$_2$O. The precipitate was then filtered and washed with Et$_2$O (3 mL) and dried under vacuum. Yield: 157 mg, 85 %.

**Method 2.** A suspension of Ni(OTf)$_2$ in DCM/MeCN was added to a DCM solution of NCHS2 (25 mg, 0.0914 mmol). The solution was stirred and heated to 70 °C overnight, after which a green solution formed. The solution was dried *in vacuo* and washed with pentane and diethyl ether. X-ray quality crystals were obtained by slow pentane diffusion into a dichloromethane solution at room temperature. Yield: 35 mg, 65%. ¹H NMR (CD$_3$CN, 500 MHz), δ (ppm): 7.32 (t, 1H, Ar), 7.25 (s, 1H, Ar), 6.96-6.87 (m, 5G, Ar), 3.96 (s, 4H, CH$_2$), 3.86 (s, 4H, CH$_2$).

UV−vis (MeCN; λ nm (ε, M⁻¹ cm⁻¹)): 273 (233)

Elemental analysis: found, C 31.67, H 2.77 N 4.84%; calculated C$_{17}$H$_{16}$F$_6$N$_2$NiO$_6$S$_4$, C 31.64, H 2.50, N 4.34%.

MALDI-TOF MS (*m/z*): 479.9156, calcd. for [(NCHS2)NiOTf]⁺, C$_{16}$H$_{15}$F$_3$NNiO$_3$S$_3$: 479.9520.

Evan's Method: μ$_{eff}$ = 1.30 μ$_B$

### Preparation of [(NCS2)Ni(MeCN)$_2$][PF$_6$], 2

To a DCM solution of NCBrS2 (30 mg, 0.087 mmol), a suspension of Ni(COD)$_2$ (24 mg, 0.087 mmol) was added. The mixture was stirred at room temperature for 5 min. The solution was filtered through celite to remove Ni black. The desired product was crashed out as a reddish brown powder by adding an excess amount of Et$_2$O. Yield: 13 mg, 54%. X-ray quality crystals of [(NCS2)Ni(μ-Br)]$_2$ were obtained by slow diffusion of Et$_2$O diffusion into a dichloromethane solution at –35 °C

UV−vis (DCM; λ nm (ε, M⁻¹ cm⁻¹)): 335, 470 nm.

Elemental Analysis: found, C 41.35%, H 3.15%, N 3.19%; calculated C$_{30}$H$_{28}$Br$_2$N$_2$Ni$_2$S$_4$·CH$_2$Cl$_2$, C 41.06, H 3.33, N 3.09%.

MALDI-TOF MS (m/z): 820.8760; calcd. for [(NCS2)Ni(μ-Br)]$_2$ + H⁺, C$_{30}$H$_{28}$Br$_2$N$_2$Ni$_2$S$_4$: 820.8267.

To a DCM solution of [(NCS2)Ni(μ-Br)]$_2$, TlPF$_6$ was added in a MeCN solution. The color changes from red to yellow-brown. It was stirred for two hours, and the solution was filtered through a syringe filter to separate the TlBr. The dark brown solution was reduced in volume and a brown powder was crashed out by adding excess diethyl ether. Despite several attempts using different counterions, no crystal of 2 could be grown. But it was characterized by IR, ESI-MS, EA, and UV–vis. A solution magnetic moment of 2.68 μ$_B$, corresponding to two unpaired electrons is suggestive of an octahedral geometry in solution.

IR (ν$_{CN}$): 2331, 2303 cm⁻¹

UV−vis (MeCN; λ nm (ε, M⁻¹ cm⁻¹)): 273 (280), 308 (60)

ESI-MS (*m/z*): 518.9871; calcd for [(NCS2)Ni(MeCN)(OTf)]-H, C$_{18}$H$_{16}$F$_3$N$_2$NiO$_3$S$_3$: *m/z* 518.9629.

Evan's Method: μ$_{eff}$ = 2.68 μ$_B$

When attempting to recrystallize 2 out of a MeCN/Et$_2$O solution, we often found crystals of [(NCHS2)Tl][PF$_6$], which points towards the proclivity of the soft thioether ligands to coordinate to the soft Tl⁺.

Elemental Analysis: found, C 38.38%, H 3.38%, N 6.46%; calculated C19H20F6N3NiPS2·0.1 TlPF$_6$, C 38.48, H 3.40, N 7.08%.

### Preparation of [(NCS2)Ni(Br)(MeCN)][OTf], [2-Br]⁺

To a solution of [(NCS2)Ni(μ-Br)]$_2$ (30 mg, 0.037 mmol) in DCM, an MeCN solution of AgOTf (9.34 mg, 0.037 mmol) was added. The solution was covered with Al-foil to prevent the photodecomposition of AgBr. The solution was filtered through a syringe filter to separate AgBr and a golden-brown solution was obtained. The solution was reduced in volume and a brown powder was crashed out with diethyl ether (91%). X-ray quality crystals of [2-Br] were grown by the vapor diffusion of diethyl ether into an acetonitrile solution of [2-Br] at −35 °C

IR: 2323, 2295 cm⁻¹ (ν$_{CN}$)

Evan's method: μ$_{eff}$ = 1.79 μ$_B$

Elemental analysis: found, C 30.62%, H 2.32%, N 3.63%; calculated C$_{18}$H$_{17}$BrF$_3$N$_2$NiO$_3$S$_3$·2 CH$_2$Cl$_2$, C 31.16, H 2.75, N 3.63%.

## Preparation of [(NCS2)Ni(MeCN)$_2$][OTf]$_2$, [2]$^+$

To a solution of [(NCS2)Ni(μ-Br)]$_2$ (30 mg, 0.037 mmol) in DCM, an MeCN solution of AgOTf (19 mg, 0.08 mmol) was added. The solution was covered with Al-foil to prevent the photodecomposition of AgBr. The solution was filtered through a syringe filter to separate AgBr and a pink solution was obtained. The solution was reduced in volume and a red powder was crashed out with diethyl ether (87%). X-ray quality crystals of **[2]$^+$** were grown by the vapor diffusion of diethyl ether into an acetonitrile solution of **[2]$^+$** at −35 °C.

IR: 2324, 2292 cm$^{-1}$ ($\nu_{CN}$)

UV–vis (MeCN; λ nm (ε, M$^{-1}$ cm$^{-1}$)): 513 (267), 720 (100)

Evan's method: $\mu_{eff}$ = 1.77 $\mu_B$

Elemental analysis: found, C 32.27%, H 3%, N 5.98%; calculated C19H20F12N3NiP2S2, C 32.46%, H 2.87%, N 6.30%.

## Data availability

Crystallographic data for compounds NCHS2, **1**, **2-Br**, **[2]$^+$** and **[2-Br]$^+$** are available free of charge from the Cambridge Crystallographic Data Centre (CCDC) under deposition numbers 2053834, 2053836, 2053837, 2195927 and 2195932, respectively. All data generated and analyzed are included in the Supplementary Information, including synthetic details, spectroscopic and electrochemical characterization of new compounds, electrocatalysis, mechanistic studies, computational details, and X-ray crystallographic details. The bond distances and angles for NCHS2 and **1** are included as Supplementary Data Files 1 and 2, respectively.

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

## Acknowledgements

We thank the National Science Foundation (CHE-1925751 and CHE-2155160 to L.M.M.) for financial support, and the Department of Energy BES Catalysis Science Program (DE-SC0006862 to L.M.M.) for support of the initial ligand synthesis work. We thank Prof. Nigam P. Rath (Univ. of Missouri—St. Louis) for obtaining the crystal structures of NCHS2 and **2**. We thank Dr. Toby Woods (University of Illinois at Urbana-Champaign) for his help with the solution of crystal structure data for **[2-Br]⁺**. We also thank Dr. Leonel Griego and Yusuff Moshood for helpful discussions regarding EPR simulations. We thank Qi Hua (Gewirth group) for her assistance in collecting the SEM/EDX data. We thank the operators of the research facilities at the University of Illinois at Urbana-Champaign for their help.

## Author contributions

L.M.M. conceived the overall project. S.C., S.S., G.T., and L.M.M. conceived and designed the experiments, S.C., S.S., G.T. carried out the experimental work, H.N. obtained and analyzed the data for the crystal structure of **2-Br**. S.C. and L.M.M. performed computational calculations, and S.C., G.T., S.S., L.M.M. analyzed the computational and experimental data. All authors assisted in writing the manuscript.

## Competing interests

The authors declare no competing interests.
