## [Peer Review File · Nature Communications]

Title – Characterization of Paramagnetic States in an Organometallic Nickel Hydrogen Evolution Electrocatalyst

Journal – Nature Communications

Summary

The authors report a mononuclear nickel complex featuring a NCHS2 ligand which they claim to be an active electrocatalyst toward the hydrogen evolution reaction, with reported turnover frequencies of 3000 s^{-1} . This species operating through a previously unobserved mechanism involving reversible CH activation of the ligand, leading to an inverse isotope effect for HER. Due to the features of the ligand, the authors claim this is an example of operating through the “elusive Ni^{I/III} cycle” by way of the mix of soft and hard donors which stabilize the Ni^I and Ni^{III} states. A Ni^{I/III} redox cycle would be a more accurate model of the Ni-L and Ni-C states in the [NiFe]H₂-ase.

Pros –

- 1) The synthesis is beautiful. The ligand design is inspired. The nickel complexes are well presented.
- 2) The text was in general well written. The engagement of a possibility of proximity effect of a C-H bond within the ligand in order to mix soft and hard donors to tune the states that are stable is a great one. It is also an important pursuit to more accurately model the Ni states.
- 3) We agree with the authors’ statement: “Switching between redox states by via a C-H bond activation, while commonly proposed for organometallic reactivity, is a new paradigm in small molecule electrocatalysis. “ They are to be commended for this exciting design principle that is displayed in their nickel complex.

Cons –

- 1) Referring to the final statement above: They are INDEED to be commended for this exciting design principle that is displayed in their nickel complex. At several points, particularly the H/D exchange and kinetic isotope effect into the proximal C-H bond of the ligand, a more detailed description would be preferable, rather than just sandwiching in important key data. That is, the manuscript could be much clearer in this area of data presentation—probably if split into two publications.
- 2) While the authors do a reasonable job of convincing the reviewer of the importance of this story with respect to odd oxidation states of Ni as seem to be necessary in the catalytic cycle (rather than the off-cycle oxygen damaged species of Ni-Fe hydrogenase) it is not clear whether or not the authors mean to invoke C-H bond activation as a viable process in

the enzyme itself. Perhaps this reviewer missed a key sentence in this regard, however it should be made so clear that missing it would be impossible.

- 3) The electrochemistry is not established with certainty. The authors failed to properly test for surface adsorbed decomposition products. The presence of the catalyst actually seems to inhibit HER from the background electrode reduction as shown in their own data. Since many of the following experiments are guided from the electrochemistry results, the interpretation of faulty electrochemical data might lead to incorrect mechanistic understanding.
- 4) The manuscript was so nicely written that this misleading data was buried. Indeed there is an abundance of “see SI” statements that hide the uncertainties.

Conclusion; overall remarks:

I do not believe this manuscript to be ready for publication, certainly not in the prestigious Nature Communications. The authors might consider excising the synthetic and structural data and not rely on the HER catalysis as the *raison d’etre* .

Further comments/Edits

- 1) Pg. 13 – Correct unneeded “Fig.” in the line “... with this ligand framework. Fig. σ -bond metathesis of B-H and Si-H”
- 2) **Electrochemistry** –
 - a. All CV’s in the manuscript and SI seem to start around -0.1 V vs $Fc^{+/0}$ which are concurrent with the -0.3 V oxidation feature. What is the open circuit potential of the process? Is the -0.3 V feature there if the potential before the reductive event at -1.5 V? is reversed In other words are they actually linked features as suggested in the manuscript. As depicted, it seems like the experiments are starting in the middle of an oxidative event.
 - b. Fig S24 – How is this complex an electrocatalyst when proton reduction from TFA is initiated at exactly the same place and appears to pass more current without the catalyst.
 - c. Fig S25 – Again, that is the potential that proton reduction from AcOH commences on a bare GC electrode.
 - d. Fig S32 – Along with the next point, this suggests that there is deposition on the electrode surface, and the process is not a homogeneous as interpreted and reported.
 - e. Fig S41 & S42 – “To check for the catalytic activity of any surface-adsorbed species, the working electrode was dipped into the catalyst-containing solution before being rinsed with excess MeCN and placed into a fresh solution of MeCN containing 0.1 M TBAPF₆ and 43 mM TFA.” I do not think this is an accurate application of the “rinse test” and its use. Of course if you simply dip the electrode and rinse it off, nothing will be on the surface. The “rinse test” should be post-electrolysis, whereupon the electrode is removed and rinsed off without

polishing, and placed into another substrate-containing solution without catalyst. Unless this reviewer is not understanding the description of the researchers, this inaccurate application of the “rinse test” calls into question all of the bulk electrolysis experiments as the current passed could be from decomposition into surface adsorbed species which are frequently the real catalyst.

- f. There are other issues, but they all stem from the major issues highlighted above.

The manuscript reported by L. M. Mirica and co. describes a new Ni electrocatalyst which is operative through a Ni(I/III) cycle. Moreover, the authors identified and characterized largely elusive intermediates of great interest, with some reminiscence to hydrogenase and have the potential to inspire new mixed hard-soft ligands in organometallic complexes for small molecule activation. For these reasons, the manuscript deserves to be published in a broad audience journal.

However, there are some considerations to consider before publication.

- A) In the abstract, the authors describe the overpotential as moderate. However, it may be better to present the overpotential and let the readers infer themselves.
- B) In the introduction, the production of H₂ is produced in electrolyzers. It is not clear why to introduce fuel cells. Maybe the authors want to change the first sentences of the introduction.
- C) In several places appears, the notation NiOTf₂, but it is more confusing than the suggested Ni(OTf)₂.
- D) Regarding the reactivity in synthesizing the different species, some reactions could bring light to the reactivity but are not discussed. For example, maybe authors already considered or tried the reactions as follows,
- The reaction of the NCHS₂ with Ni(0)(COD)₂ should not be an alternative to obtaining the Ni(I) and the Ni(III)-H complexes.
 - The reaction of NDCHS₂ with Ni(II) and the resulting complex reacted with CoCp*₂ to analyze EPRs of Ni(I) and Ni(III)-D. Therefore, the equilibrium analysis could give information about the Ni(I)/Ni(III)-D thermodynamics versus the KIE obtained via CV.
 - It could be possible to shift the Ni(III)-H to Ni(III)-Br by reaction with HBr. The idea here is to obtain the same complex as [2-Br]⁺. Alternatively, the equivalent Ni(III)-Br₂ should be possible to obtain via the two routes. The main reason is that the Ni(III)-H is not fully characterized, and additional reactivity will establish more robust conclusions.
- E) It is not discussed why the CV peaks of the Ni(II/I) redox wave are far apart (i-peak(red) -1.5 and i-peak(oxd) -0.26 V). However, reasoning this unusual behaviour is important.
- F) The DFT scheme of Fig 6 should be substantially improved. It is advised to separate the profiles in triplet and singlet, but with the same speciation, there are also coordination changes. Moreover, the energies seem that do not consider redox. Including the catalytic redox will become more evident if the catalytic pathway is DFT allowed. Moreover, illustration of alternative profiles should be included in the SI.
- G) Could it be possible that the S atoms are protonated? Could it be excluded based on the KIE?
- H) Which is the KIE of the first peak? Could it be possible that the first catalytic peak involves the 1-3-5 catalytic cycle, while at higher overpotentials, the second catalytic peak, the Ni(III)-H is then reduced to Ni(II)-H activating the 1-3-4-5 path?
- I) Should be the KIE calculated at the plateau, considering the used equations?

I enjoyed reading the manuscript entitled '*Characterization of Paramagnetic States in an Organometallic Nickel Hydrogen Evolution Electrocatalyst*'. It is a well-studied and nicely presented report. The mechanism of the hydrogen evolution reaction is well supported by both the experiments and theoretical calculations.

My only concern is the novelty of this report and its suitability for *Nature Communication*. Although the system showed a high TOF, the overpotential is not that great (670 mV) and that is also in presence of trifluoroacetic acid.

A few minor comments on the characterizations are as follows:

For (NCHS₂)Ni(OTf)₂

ESI-MS (m/z): 653.0336; calcd for [(NCHS₂)Ni(OTf)₂]+H+Na, C₁₇H₁₆F₆NNaNiO₆S₄: m/z 652.9016. The difference is more than the usually accepted value (according to my standard). Please provide the isotopic distribution pattern of the mass envelope.

For [(NCS₂)Ni(MeCN)₂][PF₆]

Elemental Analysis: found, C 41.35%, H 3.15%, N 3.19%; calculated C₃₀H₂₈Br₂N₂Ni₂S₄·**CH₂Cl₂**, C 41.06, H 3.33, N 3.09%.

ESI-MS/HRMS analysis is missing, and elemental analysis was calculated including CH₂Cl₂, a low boiling solvent.

Please provide the isotopic distribution pattern of all the mass envelopes.

UNIVERSITY OF ILLINOIS
AT URBANA-CHAMPAIGN

DEPARTMENT OF CHEMISTRY
Box 49-6, A442 CLSL
600 South Mathews Avenue
Urbana, IL 61801-3364

LIVIU M. MIRICA
William H. & Janet G. Lycan Professor of Chemistry
Telephone: +1-217-300-1939
mirica@illinois.edu

Point-by-point response to reviewers' comments (in italics):

Reviewer 1 (Comments to the authors):

The authors report a mononuclear nickel complex featuring a NCHS2 ligand which they claim to be an active electrocatalyst toward the hydrogen evolution reaction, with reported turnover frequencies of 3000 s⁻¹. This species operating through a previously unobserved mechanism involving reversible CH activation of the ligand, leading to an inverse isotope effect for HER. Due to the features of the ligand, the authors claim this is an example of operating through the “elusive NiI/III cycle” by way of the mix of soft and hard donors which stabilize the NiI and NiIII states. A NiI/III redox cycle would be a more accurate model of the Ni-L and Ni-C states in the [NiFe]H₂-ase.

Pros –

- 1) The synthesis is beautiful. The ligand design is inspired. The nickel complexes are well presented.
- 2) The text was in general well written. The engagement of a possibility of proximity effect of a C-H bond within the ligand in order to mix soft and hard donors to tune the states that are stable is a great one. It is also an important pursuit to more accurately model the Ni states.
- 3) We agree with the authors' statement: “Switching between redox states by via a C-H bond activation, while commonly proposed for organometallic reactivity, is a new paradigm in small molecule electrocatalysis. “ They are to be commended for this exciting design principle that is displayed in their nickel complex.

Our Response. *We thank the reviewer for their kind comments.*

Cons –

- 1) Referring to the final statement above: They are INDEED to be commended for this exciting design principle that is displayed in their nickel complex. At several points, particularly the H/D exchange and kinetic isotope effect into the proximal C-H bond of the ligand, a more detailed description would be preferable, rather than just sandwiching in important key data. That is, the manuscript could be much clearer in this area of data presentation—probably if split into two publications.

Our Response. *We thank the reviewer for their comment. We have added in the main text an expanded discussion of the H/D exchange as seen in the NMR, as well as an expanded discussion of the observed electrochemical KIE.*

2) While the authors do a reasonable job of convincing the reviewer of the importance of this story with respect to odd oxidation states of Ni as seem to be necessary in the catalytic cycle (rather than the off-cycle oxygen damaged species of Ni-Fe hydrogenase) it is not clear whether or not the authors mean to invoke C-H bond activation as a viable process in the enzyme itself. Perhaps this reviewer missed a key sentence in this regard, however it should be made so clear that missing it would be impossible.

Our Response. *We thank the reviewer for pointing out this ambiguity in the writing of the text. We do not propose a C-H activation process in the enzyme. We sought to highlight that redox switching between Ni^I and Ni^{III}, which is a key feature of the enzyme, is also seen in our catalyst, albeit via a C-H activation process, instead of protonation of the Ni^I center. We have added this clarification in the manuscript (Page 14).*

3) The electrochemistry is not established with certainty. The authors failed to properly test for surface adsorbed decomposition products. The presence of the catalyst actually seems to inhibit HER from the background electrode reduction as shown in their own data. Since many of the following experiments are guided from the electrochemistry results, the interpretation of faulty electrochemical data might lead to incorrect mechanistic understanding.

Our Response. *We thank the reviewer for the comment. We acknowledge that in the previous version of the manuscript, while the homogeneous nature of the catalyst was examined by a rinse test and linearity of i_{pc} vs \log (scan rate), it may not have been established with certainty. We do think that the additional electrochemical experiments and analyses have been performed according to the required standards. We have performed several control experiments to test for heterogeneity (vide infra) and have updated the manuscript and SI to reflect the same.*

4) The manuscript was so nicely written that this misleading data was buried. Indeed, there is an abundance of “see SI” statements that hide the uncertainties.

Our Response. *We apologize for the lack of clarity and transparency. We agree with the reviewer’s previous comments that the catalytic performance of the complex is not the crux of the story told here. We wanted to highlight the mechanistic implications of engaging a proximal C-H bond in hydrogen evolution, and its parallels to the paramagnetic hydrogen evolution pathway followed by [NiFe] hydrogenases. Hence, we decided to include the bulk of the electrochemical catalysis data as a part of the Supplementary Information, and not in the main text. Our intention was NOT to hide any uncertainties. We believe that we have addressed the concerns the raised by the reviewer, especially related to the tests of homogeneity, and have modified the manuscript and SI to reflect the same.*

Conclusion; overall remarks:

I do not believe this manuscript to be ready for publication, certainly not in the prestigious Nature Communications. The authors might consider excising the synthetic and structural data and not rely on the HER catalysis as the *raison d'être*.

Our response. *We thank the reviewer for their comment. We agree with the reviewer that the HER catalysis is not the *raison d'être* of this study. The catalytic parameters for the Ni compound described in the study compares favorably with related systems, but is definitely not the fastest or most energy efficient. However, we do believe that the concept of involving a proximal C-H bond to form organometallic intermediates in hydrogen evolution maps an organometallic mechanism on to a reaction that is deemed vital for alternative energy. Hence, we do not believe that separating the synthetic and electrochemical data is necessary. We have, updated the manuscript with an expanded discussion on C-H activation and inverse KIE. We hope all the changes and additional experiments performed will make this study worthy of being considered for publication in Nature Communications.*

Further Comments/Edits

1) Pg. 13 – Correct unneeded “Fig.” in the line “.... with this ligand framework. Fig. σ -bond metathesis of B-H and Si-H”

Our Response. *We apologize for the typo. We have made the necessary corrections in the manuscript.*

2) Electrochemistry –

a. All CV's in the manuscript and SI seem to start around -0.1 V vs Fc^{+/0} which are concurrent with the -0.3 V oxidation feature. What is the open circuit potential of the process? Is the -0.3 V feature there if the potential before the reductive event at -1.5 V? is reversed In other words are they actually linked features as suggested in the manuscript. As depicted, it seems like the experiments are starting in the middle of an oxidative event.

Our Response. *We thank the reviewer for this comment. The open circuit potential of the process is -0.35 V vs Fc^{+/0}. The oxidative feature at -0.3 V is absent if the scan is reversed at -0.7 V (onset of the reductive process at -1.5 V). This indicates that the features are indeed linked. In addition, a CV was recorded oxidatively first to reiterate that the features are linked. On scanning oxidatively up to +0.1 V, no Faradaic process is seen. The oxidative feature at -0.3 V is seen when the scan is continued up to -1.75 V.*

b. Fig S24 – How is this complex an electrocatalyst when proton reduction from TFA is initiated at exactly the same place and appears to pass more current without the catalyst.

Our Response. We apologize for the confusion created by Fig, S24. We have removed the figure and would like to point the reviewer towards Fig, S23, where we have shown that the window of hydrogen evolution with the catalyst is different from that of the bare glassy carbon. Both the onset, E_{pc1} , and E_{pc2} are well separated from the potential of hydrogen evolution at bare glassy carbon. As seen in the controlled potential electrolysis data, the potentials chosen for electrolysis were between E_{pc1} and E_{pc2} , where the contribution of glassy carbon towards hydrogen evolution current density is significantly less. We do acknowledge that the performance is not remarkably better than the glassy carbon electrode. Current research efforts in the lab are underway to develop better catalysts based on the mechanistic insights gained from this study.

We also note that the catalytic performance of other functional models of [NiFe] hydrogenase is not very different from the glassy carbon, especially in presence of strong acids, as exemplified by the model compound reported by Duboc and coworkers (Nature Chem, 2016, 8, 1054–1060).

c. Fig S25 – Again, that is the potential that proton reduction from AcOH commences on a bare GC electrode.

Our Response. *We agree with the reviewer’s comments and have removed the data involving the use of acetic acid as the proton source from the paper and SI, as it does not add anything new to the science discussed. We are currently examining the effect of using weaker and stronger acids for hydrogen evolution with modified versions of this catalyst.*

d. Fig S32 – Along with the next point, this suggests that there is deposition on the electrode surface, and the process is not a homogeneous as interpreted and reported.

Our Response. *We thank the reviewer for the comment. We would like to point out that this is the CV of compound 2, which is not the main catalyst discussed in the paper. In fact, 2 (and 2-Br) are air-sensitive and it is reasonable to assume some decomposition occurs in solution. We acknowledge the importance of ensuring homogeneity in solution, and we have performed several additional experiments to further support the homogeneous nature of our HER catalyst, as described below (comment e).*

e. Fig S41 & S42 – “To check for the catalytic activity of any surface-adsorbed species, the working electrode was dipped into the catalyst-containing solution before being rinsed with excess MeCN and placed into a fresh solution of MeCN containing 0.1 M TBAPF6 and 43 mM TFA.” I do not think this is an accurate application of the “rinse test” and its use. Of course, if you simply dip the electrode and rinse it off, nothing will be on the surface. The “rinse test” should be post-electrolysis, whereupon the electrode is removed and rinsed off without polishing, and placed into another substrate-containing solution without catalyst. Unless this reviewer is not understanding the description of the researchers, this inaccurate application of the “rinse test” calls into question all of the bulk electrolysis experiments as the current passed could be from decomposition into surface adsorbed species which are frequently the real catalyst.

Our Response. *We thank the reviewer for their insightful comments. We would like to point out that the rinse test procedure described in the Supporting Information was adapted from a recent report by Mayer and coworkers (Sci. Adv., 2020, 6, eaaz3318). We also performed the rinse test as described by the reviewer. A cyclic voltammogram was recorded with 1 mM catalyst in a N₂-saturated 43 mM TFA/MeCN solution (black trace). Electrolysis was then performed with a glassy carbon electrode at -1.7 V vs Fc⁺⁰ for 1 hour. The electrode was then rinsed with acetone and acetonitrile. A CV was then recorded under nitrogen (red trace), which showed no discernible Faradaic features. With 43 mM TFA added, a voltammogram was seen that resembles the voltammogram seen for bare glassy carbon, suggesting*

minimal contribution from surface adsorbed species. The current and potential windows are higher and more anodic respectively for the catalyst, as seen in the figure below.

In addition to the rinse test described above, we also performed SEM and EDX analysis of a glassy carbon disk electrode after the rinse test. Electrolysis was performed for 30 minutes at $-1.7\text{ V vs Fc}^{+/0}$ (E_{pc2}) with the electrode as described above and rinsed with acetone and acetonitrile. The CV and bulk electrolysis traces are shown below:

The glassy carbon disk was then rinsed with acetonitrile and used for analysis. The results of the SEM and EDX show that there is negligible Ni deposition after rinsing off with organic solvents. The supplementary information has been updated with all the results of the surface analysis.

Scanning electron micrograph of the surface of the glassy carbon electrode after rinse:

Tabulated results EDX analysis of the surface of the glassy carbon after rinse test:

Element	Atomic %	Atomic % Error	Weight %	Weight % Error	Net Counts
C	91.1	0.5	89.6	0.5	37 785
N	8.9	0.7	10.2	0.8	316
O	0.0	---	0.0	---	0
S	0.0	0.0	0.1	0.1	839
Ni	0.0	0.0	0.2	0.2	22

Full details of the results obtained from SEM and EDX analyses, including zoomed-in micrographs of the glassy carbon surface, have been included in the Supplementary Information.

f. There are other issues, but they all stem from the major issues highlighted above.

Our Response. We hope that we have addressed the major issues with the electrochemistry, as highlighted by the reviewer and hope that this satisfies their concerns with the study. We appreciate the comments and suggestions, as we believe the comments have added to the rigor of the work discussed.

Reviewer 2 (Comments to the authors):

The manuscript reported by L. M. Mirica and co. describes a new Ni electrocatalyst which is operative through a Ni(I/III) cycle. Moreover, the authors identified and characterized largely elusive intermediates of great interest, with some reminiscence to hydrogenase and have the potential to inspire new mixed hard-soft ligands in organometallic complexes for small molecule activation. For these reasons, the manuscript deserves to be published in a broad audience journal.

Our Response. *We thank the reviewer for their kind comments.*

However, there are some considerations to consider before publication.

A) In the abstract, the authors describe the overpotential as moderate. However, it may be better to present the overpotential and let the readers infer themselves.

Our Response. *We have removed the word 'moderate' from the abstract.*

B) In the introduction, the production of H₂ is produced in electrolyzers. It is not clear why to introduce fuel cells. Maybe the authors want to change the first sentences of the introduction.

Our Response. *We thank the reviewer for pointing this out and have made the necessary changes to the manuscript.*

C) In several places appears, the notation NiOTf₂, but it is more confusing than the suggested Ni(OTf)₂.

Our Response. *We thank the reviewer for the comment, and we have changed the notation from NiOTf₂ to Ni(OTf)₂ in all places in the manuscript.*

D) Regarding the reactivity in synthesizing the different species, some reactions could bring light to the reactivity but are not discussed. For example, maybe authors already considered or tried the reactions as follows,

- The reaction of the NCHS₂ with Ni(0)(COD)₂ should not be an alternative to obtaining the Ni(I) and the Ni(III)-H complexes.

Our Response. *We thank the reviewer for their suggestion. We did try a reaction with NCHS₂ and Ni(COD)₂ in DCM at -35 °C, but ended up getting an intractable mixture of the ligand and Ni black. We did observe a signal at -29 ppm, which could be putative Ni^{II}-H species. But all attempts to isolate the compound were unsuccessful. We anticipate that we need stronger donors to stabilize the compound, as the only reported Ni^{II}-H generated by the oxidative addition of Ni⁰ into a phenyl C-H bond, reported by Fout and coworkers (*Organometallics* **2015**, 34, 399–407) had strongly donating NHC ligands.*

- The reaction of NDCHS₂ with Ni(II) and the resulting complex reacted with CoCp*₂ to analyze EPRs of Ni(I) and Ni(III)-D. Therefore, the equilibrium analysis could give information about the Ni(I)/Ni(III)-D thermodynamics versus the KIE obtained via CV.

Our Response. We would like to direct the reviewers to pages S55 to S57 in the Supporting Information where we have compared the reactivity of the Ni^{II} complexes of the NCHS2 and NCDS2 ligands towards one electron reduction with cobaltocene. While we do see subtle differences, it can be hard to conclude about the equilibrium from the EPR data alone. It has been shown previously (Peters et al., J. Am. Chem. Soc. 2020, 142, 17, 7827–7835, Ref. 24) that CW-EPR is not enough to differentiate between a Ni^{III}-H and Ni^{III}-D. We have added a discussion about the same in the main text. Moreover, we reason that probing this equilibrium will be more facile by changing ligand electronics, which is a current research direction in our lab.

- It could be possible to shift the Ni(III)-H to Ni(III)-Br by reaction with HBr. The idea here is to obtain the same complex as [2-Br]⁺. Alternatively, the equivalent Ni(III)-Br₂ should be possible to obtain via the two routes. The main reason is that the Ni(III)-H is not fully characterized, and additional reactivity will establish more robust conclusions.

Our Response. We appreciate the reviewer's suggestion, but we think that adding a strong mineral acid to the complex will cleave the Ni-C bond and cause demetallation. We do acknowledge that the Ni^{III}-H is not fully characterized, and we are looking at ways to generate sufficient amounts of the Ni^{III}-H by tuning the ligand, in order to be able to characterize it. Please keep in mind that, since this Ni^{III}-H species **1** is proposed to be catalytically active in the current ligand system, the detailed characterization of this species is very unlikely. Our future approaches focus on designing analogs that are less catalytically active, and thus should lead to more stable intermediate species.

E) It is not discussed why the CV peaks of the Ni(II/I) redox wave are far apart (i-peak(red) -1.5 and i-peak(oxd) -0.26 V). However, reasoning this unusual behaviour is important.

Our Response. We thank the reviewer for the comment. We posit that the significant chemical changes that take place upon the one-electron reduction could be responsible for the large peak separation of the cathodic and anodic waves for **1**. The chemical reduction experiments suggest that **3** and **4** are in equilibrium and they are very different chemical species – **3** is a Ni^I with no C-Ni bond, while **4** is a C-H activated Ni^{III} intermediate. As it can also be seen in Fig. S32, in the CV of [2]⁺ the Ni^{II/III} oxidation appears at a very similar potential. Hence, we hypothesize that **4** can get reduced to a C-H activated Ni^{II} species at those cathodic potentials and the anodic wave observed is due to the oxidation of the said Ni^{II} species. As with many aspects of this study, a comparison of the electrochemical behavior of **1** with the dipyrindine analog – [(N2S2)Ni^{II}(MeCN)₂]²⁺ is useful. As seen in Fig S37, there is no oxidative wave that is seen for the N2S2Ni complex, even though two irreversible reductive peaks are observed, which are assigned to Ni^{III/I} and Ni^{I/0} events. We have added a discussion in the manuscript about the possible reason for this electrochemical irreversibility (Pages 7 and 19).

F) The DFT scheme of Fig 6 should be substantially improved. It is advised to separate the profiles in triplet and singlet, but with the same speciation, there are also coordination changes. Moreover, the energies seem that do not consider redox. Including the catalytic redox will become more evident if the catalytic pathway is DFT allowed. Moreover, illustration of alternative profiles should be included in the SI.

Our Response. We thank the reviewer for this comment, however, we consider that the inclusion of both singlet and triplet reaction coordinates in the same DFT scheme is optimal, since it reflects the true nature

of **1**, which is shown to be a mixture of singlet and triplet states based on NMR and Evans Method. The change in the coordination number has been depicted clearly, with the loss of an acetonitrile molecule imposing an energy penalty in going from triplet to singlet.

We are also unsure what the reviewer means by “catalytic redox”. We have included the calculated catalytic reduction potential in the Figure 7 legend. We have also included the alternative profile discussed as an answer to point G in the SI (Page S122 and Fig. S92).

G) Could it be possible that the S atoms are protonated? Could it be excluded based on the KIE?

Our Response. We thank the reviewer for these questions. While it has been shown in previous instances that thiolate protonation is a possibility in molecular electrocatalysts, we do not anticipate a thioether to show the same chemistry, as it is significantly less basic, especially upon coordination to a metal center. We have included in the text (first paragraph on page 19), a sentence addressing the possibility of thioether protonation.

While it can be difficult to pinpoint such a conclusion from the KIE values, we would like to point out that for ligand centered HER in a (S₂P₂)Re complex reported by Grapperhaus et al. (J. Am. Chem. Soc. 2015, 137, 29, 9238–9241), a high KIE of 9 ± 1 was observed. The same authors have observed inverse KIEs in a separate study with a (S₂P₂)Ni complex (ref. 53), which was attributed to be consistent with “a metal hydride intermediate”, but with no explanation about its possible origin. We thus believe that our rationalization of an inverse KIE reconciles some classical results of C-H activation chemistry (refs. 50 and 51) with observations from homogeneous hydrogen evolution catalysis that have remained unexplained (refs. 52 and 53).

H) Which is the KIE of the first peak? Could it be possible that the first catalytic peak involves the 1-3-5 catalytic cycle, while at higher overpotentials, the second catalytic peak, the Ni(III)-H is then reduced to Ni(II)-H activating the 1-3-4-5 path?

Our Response. We thank the reviewer for this question. We evaluated the electrochemical KIE (average = 0.57 ± 0.2) of the first catalytic peak and the results are shown below (also included in the SI):

Run	Slope _H	Slope _D	[Slope _H /Slope _D]	[Slope _H /Slope _D] ²
1	0.138 ± 0.002	0.185 ± 0.008	0.75 ± 0.05	0.56 ± 0.2
2	0.146 ± 0.007	0.166 ± 0.002	0.88 ± 0.05	0.77 ± 0.2
3	0.136 ± 0.006	0.18 ± 0.02	0.76 ± 0.08	0.58 ± 0.3
4	0.134 ± 0.008	0.242 ± 0.008	0.55 ± 0.05	0.30 ± 0.2
5	0.136 ± 0.008	0.167 ± 0.007	0.81 ± 0.08	0.65 ± 0.3

The 1-3-5 cycle that the reviewer proposes would mean the involvement of a Ni⁰/Ni^{II}-H pathway. We think that can be precluded based on the lack of Ni⁰ feature in the CV under nitrogen. The alternative is a hydrogen atom transfer (HAT) to Ni^I, but that needs a HAT source – we have none in the system.

An inverse KIE for the first catalytic peak means, based on our hypothesis, the cleavage of a weak chemical bond to form a strong bond. So, our best guess would be a pathway which involves the

protonolysis of the $\text{Ni}^{\text{III}}\text{-H}$ to form a $\text{Ni}^{\text{III}}\text{-solvento}$ complex and evolve H_2 . The $\text{Ni}^{\text{III}}\text{-solvento}$ would then get reduced and protonated at the Ni-C bond to regenerate the resting state of the catalyst.

We also evaluated the energetics of a pathway involving protonolysis of the $\text{Ni}^{\text{III}}\text{-H}$, however the calculated barrier for such a process is very high, suggesting that it is an unlikely process. In addition, diminished FEs were observed at lower overpotentials. We have included a discussion about this in the SI and in the main text (Pages 19-20 and S121-122).

Under pure kinetic conditions (Fig. S49), the two-peak feature is replaced by a single plateau wave, which is in the high overpotential regime. It is worth noting that for the lower overpotential regime, the $E_{\text{cat}2}$ is very close to the peak from glassy carbon and could obfuscate analysis. This would also explain the higher uncertainties in the KIE.

I) Should be the KIE calculated at the plateau, considering the used equations?

Our Response. Since $(i_{\text{cat}}/i_p)^2$ is proportional to k_{cat} , we have used (i_{cat}/i_p) as a proxy for k_{cat} based on equation shown in the SI. Since current density intrinsically represents the rate of the reaction at various acid concentrations, we believe that this equation also holds for the calculation of electrochemical KIEs. While an S-shaped voltammogram or the plateau current would represent true kinetic control, differences in the current densities by titrations of H-acids and D-acids have been used previously (see Dey et al. *Inorg. Chem.* **2020**, *59*, 5292–5302; Peters et al. *Nature* **2022**, *609*, 71–76) even with non-canonical voltammograms.

Reviewer 3 (Comments to the authors):

I enjoyed reading the manuscript entitled ‘Characterization of Paramagnetic States in an Organometallic Nickel Hydrogen Evolution Electrocatalyst’. It is a well-studied and nicely presented report. The mechanism of the hydrogen evolution reaction is well supported by both the experiments and theoretical calculations. My only concern is the novelty of this report and its suitability for Nature Communication. Although the system showed a high TOF, the overpotential is not that great (670 mV) and that is also in presence of trifluoroacetic acid.

Our Response. *We thank the reviewer for their kind comments. We would like to state that this study was focused on unraveling a new mechanism in homogeneous hydrogen evolution catalysis involving C-H activation operable in an organometallic hydrogen evolution catalyst, and not necessarily developing the best-performing catalyst.*

As for the overpotential, we would like to point out both the onset potential, E_{pc1} and E_{pc2} are well separated from the potential of hydrogen evolution at bare glassy carbon using the same acid (Fig S23). As shown in an updated rinse test (Figs. S41-44), the catalyst is homogeneous and displays anodic potentials and higher current densities than the post-rinse glassy carbon electrode. The performance also compares favorably to some of the reported functional and structural models of [NiFe] hydrogenases (see text), where Ni^I has been proposed to be the active state. This certainly does not outcompete the fastest homogeneous Ni-based catalysts (like the DuBois N2P2 systems), which often activate the significantly more electron rich Ni^0 state towards proton reduction. A current research direction in the laboratory is to develop faster and more thermodynamically competent catalysts based on the mechanistic insights gained from studying this catalyst.

*We also note that using trifluoroacetic acid and other stronger acids has been fairly common in the homogeneous hydrogen evolution catalysis literature. As representative examples, see Rauchfuss et al. J. Am. Chem. Soc. **132**, 42, 14877–14885 (2010), where TFA was used as a proton source for a model compound of [NiFe] hydrogenase, DuBois et al. Science **333**, 863-866 (2011), where a Ni phosphine compound evolved H_2 with [DMF-H]⁺ as a proton source ($pK_a = -0.3$ in DMF). We are also looking to develop catalysts that can work with milder acids (or water), without compromising the kinetics and overpotential of HER.*

A few minor comments on the characterizations are as follows:

For (NCHS₂)Ni(OTf)₂

ESI-MS (m/z): 653.0336; calcd for [(NCHS₂)Ni(OTf)₂]+H+Na, C₁₇H₁₆F₆NNaNiO₆S₄: m/z 652.9016. The difference is more than the usually accepted value (according to my standard). Please provide the isotopic distribution pattern of the mass envelope.

Our Response. We thank the reviewer for their comment. We apologize for this oversight. Shown below is the MALDI-TOF spectrum corresponding to $[(\text{NCHS}_2)\text{NiOTf}]^+$, formed by the loss of a triflate anion to form the monocationic species. Calcd. for $\text{C}_{16}\text{H}_{15}\text{F}_3\text{NNiO}_3\text{S}_3$: m/z 479.9520.

For $[(\text{NCS}_2)\text{Ni}(\text{MeCN})_2][\text{PF}_6]$

Elemental Analysis: found, C 41.35%, H 3.15%, N 3.19%; calculated $\text{C}_{30}\text{H}_{28}\text{Br}_2\text{N}_2\text{Ni}_2\text{S}_4 \cdot \text{CH}_2\text{Cl}_2$, C 41.06, H 3.33, N 3.09%.

ESI-MS/HRMS analysis is missing, and elemental analysis was calculated including CH_2Cl_2 , a low boiling solvent.

Please provide the isotopic distribution pattern of all the mass envelopes.

Our Response. We thank the reviewer for the comment. We performed the elemental analysis of the compound $[2\text{-Br}]$ multiple times. Each time the amount of carbon found was slightly more than the expected theoretical value, which we account for by including one molecule (one per Ni) or ‘half’ a molecule (one every two Ni centers) of dichloromethane. We find that this is chemically reasonable as dichloromethane was used as a solvent in the synthesis of $[2\text{-Br}]$ (i.e., the reaction of $\text{Ni}(\text{COD})_2$ with NCS_2) and can co-crystallize with $[2\text{-Br}]$. This complex is also quite insoluble in most common organic solvents, which impedes further purification by recrystallization.

Results of elemental analysis of $[2\text{-Br}]$ – four different trials are shown:

CHN Results					
Element	Theoretical	Run 13316		Run 13317	
		Found	Diff	Found	Diff
C	43.83%	42.07%	-1.76	41.92%	-1.91
H	3.43%	3.21%	-0.22	3.22%	-0.21
N	3.41%	3.2%	-0.21	3.18%	-0.23

CHN Results					
Element	Theoretical	Run 10167		Run 10168	
		Found	Diff	Found	Diff
C	43.83%	41.35%	-2.48	41.68%	-2.15
H	3.43%	3.15%	-0.28	3.19%	-0.24
N	3.41%	3.19%	-0.22	3.23%	-0.18

Calculated for $C_{30}H_{28}Br_2N_2Ni_2S_4 + 1 CH_2Cl_2$: C 42.38%, H 3.38%, N 3.24%

Calculated for $C_{30}H_{28}Br_2N_2Ni_2S_4 + 0.5 CH_2Cl_2$: C 41.06% H 3.33% N 3.09%

We also recorded the MALDI-TOF spectrum of [2-Br]. The spectrum and isotopic distribution pattern is shown below with the masses shown. The peak at $m/z = 820.8760$ corresponds to the mass of $[C_{30}H_{28}Br_2N_2Ni_2S_4+H]^+$. Calcd. for $[C_{30}H_{28}Br_2N_2Ni_2S_4+H]^+$: m/z 820.8267.

Comment 1 Chakrabarti, Sagnik
Comment 2 dctb

REVIEWERS' COMMENTS

Reviewer #1 (Remarks to the Author):

The authors have conscientiously addressed the comments of the reviewer--and made the changes requested in the instances where it was deemed necessary to increase clarity. In only one instance was there still possible ambiguity, and that involved the "rinse test". However the reviewer is ok with the manuscript as it is now, and its publication in Nature Comm. .

Reviewer #2 (Remarks to the Author):

The Authors have provided a convincing revision version of the manuscript clarifying all the questions.

Reviewer #3

Dear Editor,

I am happy with the revision provided by Prof. Liviu M. Mirica and the team. From my side, the manuscript can be accepted for publication.

With regards

Reviewers' Comments

Reviewer #1 (Remarks to the Author)

The authors have conscientiously addressed the comments of the reviewer – and made the changes requested in the instances where it was deemed necessary to increase clarity. In only one instance was there still possible ambiguity, and that involved the “rinse test”. However the reviewer is ok with the manuscript as it is now, and its publication in Nature Comm.

Our Response. We sincerely thank the reviewer for their thoughtful comments and suggestions regarding the detailed probing of the catalyst heterogeneity, as we believe it has improved the quality of the chemistry described in the paper.

Reviewer #2 (Remarks to the Author)

The authors have provided a convincing revision version of the manuscript clarifying all the questions.

Our Response. We sincerely thank the reviewer for their suggestions, as we believe it has improved the discussions and rigor of the science described in the paper.

Reviewer #3 (Letter to the Editor)

I am happy with the revision provided by Prof. Liviu M. Mirica and the team. From my side, the manuscript can be accepted for publication.

Our Response. We sincerely thank the reviewer for their suggestions, as we believe it has improved the quality of the work described in the paper.